# FedDOR: Orthogonal Initialization and Dual Regularization for Prototype Integrity in Heterogeneous Federated Learning

## Abstract

Heterogeneous Federated Learning (HFL) has garnered significant attention for its potential to leverage decentralized data while preserving privacy. One fundamental challenge in HFL is how to drastically reduce the high communication cost of transmitting model parameters. Prototype-based HFL methods have recently emerged, which exchange only class-wise representations (prototypes) among heterogeneous clients to achieve model training. However, existing methods fail to maintain the semantic integrity of prototypes during the aggregation process, compromising the global model performance in HFL. To overcome the challenge of semantic degradation in prototype aggregation, we propose a novel HFL approach termed FedDOR, which leverages Dual Orthogonal Regularization (DOR) to learn consistent and discriminative prototypes. On the client-side, our key insight is orthogonally initializing prototype embeddings to impose a maximally separated and uniformly distributed prior geometry on the feature space, providing a consistent and optimal learning target. On the server-side, DOR enforces geometric constraints to explicitly minimize intra-class variance while enlarging inter-class separation. Extensive experiments demonstrate that FedDOR achieves superior accuracy over state-of-the-art methods by significant margins, while fully preserving the communication efficiency and privacy advantages of prototype-based federated learning.

## 1 Introduction

Federated Learning (FL) has emerged as a promising paradigm for training machine learning models across decentralized data sources while preserving data privacy (Kairouz et al., 2021; McMahan et al., 2017). However, conventional FL methods like FedAvg (McMahan et al., 2017) assume homogeneous client models and data distributions (Zhang et al., 2023b; Ye et al., 2023a). This assumption is often violated in real-world scenarios due to system constraints and data collection processes (Yi et al., 2024; Tan et al., 2022b). Such heterogeneity, including both in data (non-IID) and model architectures, poses significant challenges to collaborative learning (Lu et al., 2024; Li et al., 2024; Zeng et al., 2023), leading to performance degradation and convergence issues (Li et al., 2020; Tan et al., 2022a).

To mitigate these challenges, Heterogeneous Federated Learning (HFL) has been proposed (Abdelmoniem et al., 2023), allowing clients to employ personalized FL methods without sharing raw data (T Dinh et al., 2020; Zhang et al., 2023a; Yang et al., 2023; Li et al., 2021; Collins et al., 2021). In particular, prototype-based methods (Tan et al., 2022b; Zhang et al., 2024b) have gained attention for their communication efficiency and privacy benefits (Cheng et al., 2023). These methods transfer exclusively class-specific prototypes (such as averaged feature representations) rather than full model updates, substantially reducing communication overhead and avoiding direct model leakage (Tan et al., 2022b; Zhu et al., 2021).

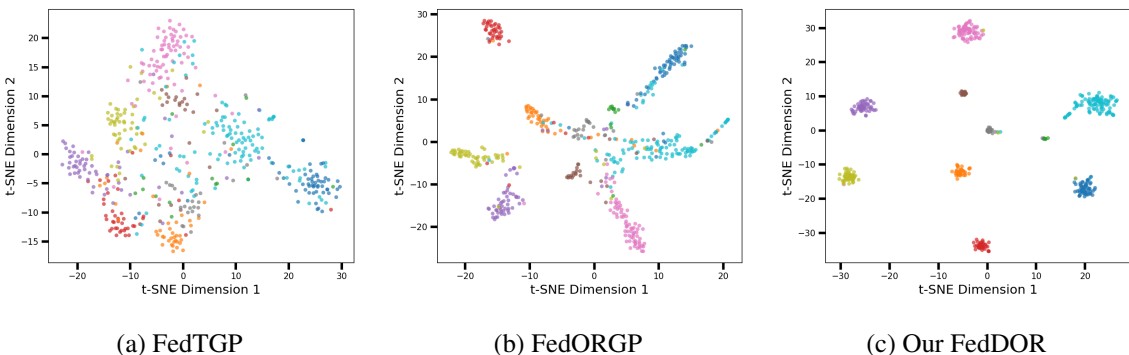

(a) FedTGP        (b) FedORGP        (c) Our FedDOR

Figure 1: The CIFAR-100 dataset is used to train the model, and then t-SNE (T-distribution random neighborhood embedding) (Maaten & Hinton, 2008) is used to visualize the classification performance of the model for unknown test samples in the feature space. Different colors represent different categories in the scatter chart.(a) Although FedTGP enhances the distinctiveness by increasing the prototype interval between classes, it still lacks sufficient distinctiveness between classes in the feature space. (b)FedORGP has improved the distinguishing ability between categories in the feature space to a certain extent, but some categories still have the feature aliasing phenomenon. (3) Under the FedDOR framework proposed by us, the original indistinguishable categories in the feature space have been significantly improved.

However, naive server-side aggregation, such as weighted averaging, undermines prototype semantics and often causes representation collapse and inter-class overlap under high heterogeneity (Zhang et al., 2024b; Dai et al., 2023). Recent fixes are partial: FedTGP (Zhang et al., 2024b) uses contrastive learning to enlarge Euclidean margins but is misaligned with the angular nature of common classification losses and still fails to fully separate some categories (Fig. 1a). FedORGP (Guo et al., 2025) enforces orthogonality as a soft constraint to encourage angular separation, yet without an explicit geometric prior it still shows feature aliasing and remains sensitive to initialization under distribution shifts (Fig. 1b).

To address these limitations, we propose FedDOR, which generates class prototypes and applies Dual Orthogonal Regularization (DOR) on the server to preserve global semantic integrity. On the client side, we use orthogonal-initialized prototype embeddings and impose uniformly distributed geometric priors. In this process, features and prototypes are L2-normalized onto the unit hypersphere, and the classifier is initialized with orthogonal weights together with a learnable scaling factor to refine decision boundaries. Intra-class orthogonality aligns samples to their prototypes for compactness, while inter-class orthogonality suppresses similarity to mismatched prototypes. Together, these losses enlarge inter-class intervals and reduce initialization sensitivity and class overlap. Figure 1(c) visualizes clearer structural consistency and larger decision margins obtained by our FedDOR under heterogeneity.

We evaluate the proposed method against several state-of-the-art HFL approaches across multiple datasets under varying settings of data heterogeneity (including both practical and Dirichlet-based non-IID partitions) and model heterogeneity (with multiple architecture groups). Experimental results demonstrate that our approach outperforms the best baseline by a significant percentage margin in accuracy under combined statistical and model heterogeneity. Our contributions can be summarized as follows.

- We propose a novel HFL framework that introduces orthogonal initialization for prototype embeddings on the client side, enforcing a maximally separated and uniformly distributed geometric prior in the feature space. Coupled with a server-side category prototype generation and refinement module, it provides consistent and discriminative learning targets across heterogeneous clients.

- We design a dual orthogonal loss mechanism that combines intra-class compactness alignment and inter-class separation enhancement, supported by L2-normalized feature and prototype projection onto a unit hypersphere and a learnable scaling factor to optimize decision boundaries.
- Extensive experiments under various heterogeneous settings demonstrate the superiority of our method in accuracy, convergence stability, and robustness compared to existing prototype-based and aggregation-based HFL methods, effectively mitigating initialization sensitivity and class overlap issues.

## 2 RELATED WORKS

**Heterogeneous Federated Learning.** Heterogeneous Federated Learning (HFL) has emerged as a critical framework addressing the challenges of collaborative learning across clients with diverse model architectures and non-IID data distributions (Zhang et al., 2023b; Ye et al., 2023a; Miao et al., 2023; Chen et al., 2024). Research in this domain has evolved along several interconnected paths, each offering distinct approaches to handle system and statistical heterogeneity (Pei et al., 2024; Lu et al., 2024). Initial investigations explored partial model sharing techniques where clients maintain private components while sharing only specific parts of their models (Huang et al., 2024; Ji et al., 2024). FedGH (Yi et al., 2023) represents this direction by learning a generalized global header to facilitate knowledge transfer across heterogeneous clients (Wang et al., 2023). While useful, these methods assume some architectural alignment and can still suffer under high data heterogeneity (Yi et al., 2024; Liao et al., 2024).

The persistent pursuit of greater architectural flexibility has led to the development of various knowledge distillation frameworks (Zhang et al., 2024a; Ye et al., 2023b). FedKD (Wu et al., 2022) employs mutual distillation between clients and servers, creating a communication-efficient alternative that avoids the need for public datasets (Wang et al., 2024b). FedGen (Zhu et al., 2021) extends this direction by leveraging generative models to extract and distribute global knowledge, though its effectiveness depends substantially on the quality of the generated samples (Feng et al., 2025). More recently, prototype-based methods have gained prominence for their communication efficiency and flexibility (Huang et al., 2023; Wang et al., 2024a; Wan et al., 2024). FedProto (Tan et al., 2022b) aggregated local class-specific prototypes via weighted averaging (Wang et al., 2025; Liao et al., 2024), FedTGP (Zhang et al., 2024b) introduced trainable global prototypes with adaptive-margin contrastive learning to improve feature discrimination (Fu et al., 2025), and FedORGP (Guo et al., 2025) incorporated orthogonality regularization for better inter-class separation. Despite these advances, current methods generally overlook the critical role of initial prototype geometry in guiding federated optimization under heterogeneous conditions.

## 3 METHOD

### 3.1 PROBLEM STATEMENT AND MOTIVATION

We consider a typical heterogeneous federated learning (HFL) system with a central server and $m$ clients, where each client $k$ has a private dataset $\mathcal{D}_k$ and a model divided into a feature extractor $f_k(\cdot; \phi_k)$ and classifier $h_k(\cdot; \theta_k)$. Following prototype-based HFL (Tan et al., 2022b), each client computes class prototypes as the mean of its features, $\mathbf{p}_k^c = \frac{1}{|\mathcal{D}k^c|} \sum (x_i, y_i) \in \mathcal{D}k^c f_k(x_i; \phi_k)$, and uploads them to the server.

The global prototype is commonly obtained by weighted averaging, $\bar{\mathbf{p}}^c = \sum k = 1^m \frac{|\mathcal{D}_k^c|}{N^c} \mathbf{p}_k^c$, where $N^c$ is the total number of class-$c$ samples. This naive aggregation has two key drawbacks. It exposes private class distribution information and produces inconsistent prototype magnitudes and directions, which lead to "prototype collapse" and reduced inter-class separability. Motivated by these limitations, we propose a method that rethinks prototype initialization and learning to ensure global prototypes remain discriminative and semantically meaningful during federated training.

### 3.2 Orthogonal Initialization of Prototype Embeddings

To overcome the drawbacks of naive aggregation, we introduce a strategic orthogonal initialization for the prototype embeddings on the client-side. Serving as a geometrically optimal prior, this initialization provides consistent and discriminative learning objectives for all heterogeneous clients, starting from the first communication round.

Let $\mathbf{W} \in \mathbb{R}^{C \times d}$ represent the matrix of prototype embeddings, where $C$ is the number of categories and $d$ is the feature dimension. Instead of initializing $\mathbf{W}$ randomly or with zeros, we constrain it to be an orthogonal matrix. Specifically, we solve the following optimization problem to maximize the minimum pairwise angular separation among the prototype vectors:

$$\max_{\mathbf{W}} \min_{i \neq j} \arccos \left( \frac{\mathbf{w}_i \cdot \mathbf{w}_j}{\|\mathbf{w}_i\| \, \|\mathbf{w}_j\|} \right) \tag{1}$$
$$\text{subject to } \|\mathbf{w}_e\|_2 = 1, \quad \forall e \in \{1, \ldots, C\},$$

where $\arccos$ denotes the inverse cosine operation. Specifically, we approximate Eq. 1 by generating points on a unit hypersphere with maximal minimum separation. This yields a set of prototypes that are approximately uniformly distributed on the unit hypersphere, such that any two distinct prototypes $\mathbf{w}_i$ and $\mathbf{w}_j$ are orthogonal or near orthogonal, satisfying $\mathbf{w}_i \cdot \mathbf{w}_j \approx 0$ for $i \neq j$.

### 3.3 Local Model Update with Global Prototype Guidance

In each local training round, clients update their local models with both their private data and the global prototypes received from the server-side. This procedure ensures that local feature representations remain semantically aligned with the global class-wise prototypes while maintaining discriminative ability.

Let $\mathcal{D}_k$ denote the local dataset of client $k$, $\mathcal{P} = \{\bar{\mathbf{p}}^c\}_{c=1}^{C}$ be the set of global prototypes. The $k$-th client's training objective combines the standard cross-entropy loss with a prototype alignment loss, formulated as:

$$\mathcal{L}_{\text{local}} = \lambda_{\text{CE}} \cdot \mathcal{L}_{\text{CE}}(h_k(f_k(x; \phi_k); \theta_k), y) + \lambda_{\text{align}} \cdot \mathcal{L}_{\text{align}}(f_k(x; \phi_k), \bar{\mathbf{p}}^y), \tag{2}$$

where $(x, y) \in \mathcal{D}_k$, $\mathcal{L}_{\text{CE}}$ as the cross-entropy loss, $\mathcal{L}_{\text{align}}$ as the global prototype alignment loss, $\lambda_{\text{CE}}$ and $\lambda_{\text{align}}$ denote the loss balancing factors to control their relative contributions.

### 3.4 Global Prototype Generation

Although the standard objective in Eq. 2 can enhance compactness and separation, the global prototype $\bar{\mathbf{p}}^y$ is insufficient to describe diverse semantic knowledge across clients. We further introduce a server-side prototype generation mechanism that produces globally consistent and well-separated class-wise representations. Specifically, the global prototypes are generated collectively through a neural transformation module by $\mathcal{P} = \mathcal{G}(\mathcal{E}; \Omega_{\mathcal{G}})$ where $\mathcal{E} = \{\mathbf{e}^c\}_{c=1}^{C} \in \mathbb{R}^{C \times d}$ represents a trainable embedding matrix containing initial prototype vectors for all $C$ categories, and $\mathcal{G}$ denotes a lightweight neural network parameterized by $\Omega_{\mathcal{G}}$, comprising two fully connected layers with ReLU activation. $\mathcal{G}$ transforms the initial embeddings into refined global prototypes that better capture the underlying class-wise characteristics across heterogeneous clients. Our proposed generation framework effectively addresses data and model heterogeneity by producing enhanced representations through neural transformation, mitigating prototype misalignment and margin shrinkage.

To further leverage the generated global prototypes for promoting feature discriminability, we introduce a dual orthogonal regularization mechanism during server-side training. Given the embedding $f(x) \in \mathbb{R}^d$ of input $x$ and refined global prototypes $\mathcal{E} = \{\mathbf{e}^c\}_{c=1}^{C} \in \mathbb{R}^{C \times d}$, we design the following two terms:

1. The intra-class alignment loss encourages feature representations to be tightly aligned with the refined global prototype of the same class. We maximize the cosine similarity between the normalized feature vector and the corresponding normalized prototype:

$$\mathcal{L}_{\text{intra}} = \frac{1}{B} \sum_{i=1}^{B} \left( 1 - \frac{f(x_i) \cdot \mathbf{e}^{y_i}}{\|f(x_i)\|_2 \times \|\mathbf{e}^{y_i}\|_2} \right), \tag{3}$$

where $B$ is the batch size and $y_i$ is the label of $x_i$. By minimizing Eq. 3, the model is encouraged to produce feature representations that cluster closely around the corresponding prototypes.

2. The inter-class orthogonality loss discourages feature representations from aligning with prototypes of incorrect classes. Concretely, we enforce inter-class orthogonality by minimizing the absolute cosine similarity between each feature vector and all non-corresponding prototypes:

$$\mathcal{L}_{\text{inter}} = \frac{1}{B} \sum_{i=1}^{B} \frac{1}{C-1} \sum_{c \neq y_i} \left| \frac{f(x_i) \cdot \mathbf{e}^c}{\|f(x_i)\|_2 \times \|\mathbf{e}^c\|_2} \right|. \tag{4}$$

Based on Eq. 4, we foster directional independence across distinct classes, thereby effectively expanding the angular margin separating different classes in the feature space.

Finally, we carry out the following optimization objective in the global prototype generation as $\mathcal{L}_{\text{intra}} + \mathcal{L}_{\text{inter}}$. By combining alignment and orthogonality constraints, $\mathcal{L}_{\text{total}}$ ensures local models learn feature representations that are semantically consistent and highly discriminative, thereby enhancing both generalization and personalization in heterogeneous federated learning.

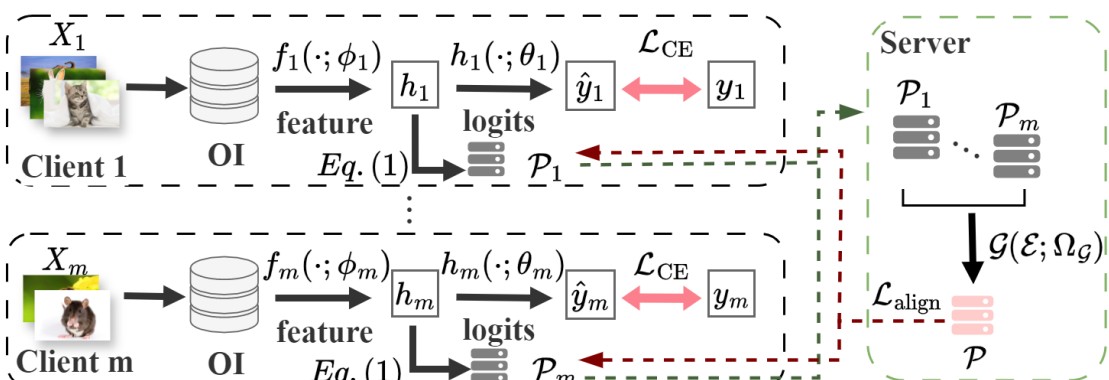

Figure 2: This figure illustrates the overall architecture of FedDOR algorithm in heterogeneous environment, which includes m clients, and each client undertakes different image recognition tasks. The local model initializes its embedded prototype by orthogonalization (OI), and iteratively updates the local prototype set by jointly optimizing cross entropy loss $\mathcal{L}_{\text{CE}}$ and consistency loss $\mathcal{L}_{\text{align}}$. Each client uploads the optimized local prototypes to the central server, and the central server uses orthogonal regularization losses $\mathcal{L}_{\text{intra}}$ and $\mathcal{L}_{\text{inter}}$ to aggregate them for generating global prototypes. Subsequently, the server broadcasts this global prototype to all clients, providing auxiliary constraints for subsequent local prototype iteration, and enhancing the consistency and generalization ability of local prototypes.

### 3.5 FedDOR Framework

Building upon the methodological components introduced above, we now present the integrated FedDOR framework. The framework employs orthogonal initialization for prototype embeddings and utilizes L2 nor-

malization to project both features and prototypes onto a unit hypersphere, enhancing inter-class separation. A learnable scaling factor is incorporated to further optimize the decision boundaries. On the server side, global semantic integrity is ensured through a prototype generation module alongside intra-class and inter-class orthogonality losses, which jointly improve feature compactness and discrimination. The complete framework operates through coordinated server-client interactions as summarized in Algorithm 1 and illustrated in Figure 2.

---

**Algorithm 1** The learning process of FedDOR

---

**Input:** Number of clients $m$, participation ratio $\gamma$, global prototypes $\mathcal{P}$ on the server, communication rounds $T$, local epochs $E$, hyperparameters $\lambda_{\text{CE}}$, $\lambda_{\text{align}}$.

**Output:** Well-trained client models $\{\theta_k\}_{k=1}^m$.

1: Each client initializes prototype embeddings using orthogonal initialization, as formalized in Eq. 1.
2: **for** iteration $t = 1, \ldots, T$ **do**
3:      Sample client subset $\mathcal{S}^t$ with $|\mathcal{S}^t| = \lceil \gamma m \rceil$.
4:      Broadcast global prototypes $\mathcal{P}$ to all clients in $\mathcal{S}^t$.
5:      **for** Client $i \in \mathcal{S}^t$ in parallel **do**
6:          Client $i$ updates its model with Eq. 2.
7:          Client $i$ calculates prototypes $\mathcal{P}_i$ by Eq. 5.
8:          Client $i$ sends $\mathcal{P}_i$ to the server.
9:      **end for**
10:      Server obtains $\mathcal{L}_{\text{intra}}$ and $\mathcal{L}_{\text{inter}}$ through Eq. 3 and Eq. 4, respectively.
11:      Server generates $\mathcal{P}$ by $\mathcal{G}(\mathcal{E}; \Omega_{\mathcal{G}})$ .
12: **end for**
13: **return** Client models.

---

Our FedDOR framework maintains the same efficient communication protocol as FedProto, and only transmits lightweight and low-dimensional class prototypes between clients and the server, which contain basic class information. This design inherently ensures that strong privacy protection and high communication efficiency can be achieved with minimum overhead in the environment with limited bandwidth. In particular, the framework eliminates the need to share any model parameters or original data, and effectively isolates sensitive local information. In addition, the process of generating these highly compressed prototypes through nonlinear mapping is irreversible in nature, which mathematically preventing potential data leakage from inversion attacks.

## 4 EXPERIMENTS

### 4.1 EXPERIMENTAL SETUP

**Datasets.** We use four widely adopted image classification datasets: CIFAR-10, CIFAR-100 (Krizhevsky et al., 2009) Flowers102 (Nilsback & Zisserman, 2008), and Tiny-ImageNet (Chrabaszcz et al., 2017). These datasets vary in complexity and number of classes, allowing us to evaluate the robustness of our method across different tasks.

**Baselines methods.** To evaluate our proposed FedDOR, we compare it with six popular methods that are applicable in HtFL, including FedGen (Zhu et al., 2021), FedKD (Wu et al., 2022), FedProto (Tan et al., 2022b), FedGH(Yi et al., 2023), FedTGP (Zhang et al., 2024b), and FedORGP (Guo et al., 2025).

**Model Heterogeneity.** To assess the robustness of all methods under model heterogeneity, we construct multiple heterogeneous model groups (MHMG) covering a wide spectrum of architectures and complexities. These groups are designed to simulate realistic federated learning scenarios with diverse client capabilities. The model configurations include: MHMG$_2$ (FedAvgCNN (Jin et al., 2010), ResNet18 (He et al., 2016)), MHMG$_3$ (ResNet10 (Zhong et al., 2017), ResNet18, ResNet34), MHMG$_4$ (FedAvgCNN, GoogleNet, MobileNet_v2 (Sandler et al., 2018), ResNet18), MHMG$_8$ (FedAvgCNN, GoogleNet (Szegedy et al., 2015), MobileNet_v2, ResNet18, ResNet34, ResNet50, ResNet101, ResNet152), and MHMG$_9$ (ResNet4, ResNet6, ResNet8, ResNet10, ResNet18, ResNet34, ResNet50, ResNet101, ResNet152). This comprehensive setup

enables thorough evaluation of each method's capability to handle varying degrees of model heterogeneity, from lightweight CNNs to very deep residual networks.

**Statistical Heterogeneity.** To simulate realistic data distribution scenarios in federated learning, we evaluate all methods under two widely-adopted non-IID data partition settings. The first setting follows a pathological non-IID partition strategy, where each client is randomly assigned a fixed number of classes from the total class set. The second setting employs a practical non-IID partition strategy using Dirichlet distribution ($\text{Dir}(\alpha)$) to simulate more nuanced and realistic label imbalance across clients. The concentration parameter $\alpha$ is set to 0.1 to create a high degree of heterogeneity, reflecting challenging and varied data distributions often encountered in real-world federated systems. These configurations allow for a comprehensive assessment of each method's robustness against statistical heterogeneity.

**Training Details.** All methods are implemented under a unified framework to ensure fair comparison. For both client and server training, we maintain a consistent batch size of 32 across all experiments. The feature dimension $K$ is set to 512 by default. The federated learning process runs for 100 communication rounds. We employ the SGD optimizer with a learning rate of 0.01 for both client and server optimization. We set $\lambda_{\text{CE}} = 10$ (weight for the classification loss), $\lambda_{\text{align}} = 1$ (weight for the prototype alignment loss) for our FedDOR on all tasks. All experiments are repeated three times, and we report the mean and standard deviation of the best test accuracy.

## 4.2 PERFORMANCE COMPARISON

Table 1: The test accuracy (%) on four datasets in the pathological and practical settings using the MHMG$_8$ model group.

| Settings | Pathological Setting | | | | Practical Setting | | | |
|---|---|---|---|---|---|---|---|---|
| Datasets | Cifar10 | Cifar100 | Flowers102 | Tiny-Imagenet | Cifar10 | Cifar100 | Flowers102 | Tiny-Imagenet |
| FedGen | 83.41±0.07 | 55.19±0.20 | 59.68±0.47 | 31.76±0.28 | 85.51±0.01 | 40.14±0.33 | 48.81±0.39 | 24.17±0.14 |
| FedKD | 82.92±0.66 | 52.42±1.24 | 53.18±1.21 | 30.92±0.91 | 84.86±1.11 | 39.25±1.43 | 44.71±0.96 | 24.66±0.64 |
| FedProto | 80.73±0.14 | 52.62±0.11 | 55.69±0.12 | 30.24±0.17 | 77.92±2.62 | 34.37±0.18 | 23.14±0.66 | 15.10±0.40 |
| FedGH | 83.84±0.13 | 54.81±0.12 | 58.56±0.23 | 31.50±0.27 | 85.76±0.18 | 39.74±0.22 | 47.23±0.71 | 23.88±0.01 |
| FedTGP | 85.07±0.19 | 53.37±0.31 | 58.40±0.57 | 29.00±0.30 | 85.20±0.16 | 38.99±0.27 | 49.61±0.41 | 22.14±0.09 |
| FedORGP | 86.09±0.31 | 58.29±0.10 | 61.50±0.32 | 30.65±0.11 | **86.98±0.34** | 40.71±0.18 | 48.74±0.22 | 21.84±0.17 |
| **Ours** | **86.28±0.06** | **63.21±0.28** | **64.76±0.52** | **36.15±0.43** | 86.20±0.08 | **47.06±0.27** | **52.39±0.71** | **28.63±0.19** |

Table 1 reports the test accuracy of all methods under both pathological and Dirichlet settings. Our method achieves the best results on most datasets and settings. Although FedORGP slightly surpasses ours on CIFAR-10 in the practical case, our approach shows clear gains on harder benchmarks. On CIFAR-100 with Dirichlet distribution ($\alpha = 0.1$) it reaches 47.06% accuracy, exceeding FedORGP by 6.35%. The improvements are also evident on Flowers102 and Tiny-ImageNet where our method achieves 64.76% and 36.15%, outperforming the strongest baselines by 3.26% and 6.18%. These results verify the effectiveness of our prototype learning framework in handling heterogeneity.

Notably, while FedTGP (Zhang et al., 2024b) is competitive in some cases, it consistently lags behind Fed-DOR, suggesting that enlarging Euclidean margins alone is insufficient. By contrast, our method achieves better semantic preservation and prototype separation, leading to more stable gains under heterogeneity.

Table 2: The test accuracy (%) on Cifar100 in the practical setting using heterogeneous feature extractors, heterogeneous classifiers, or a large number of clients ($\rho = 0.5$) with the $MHMG_8$ model group. "Res" is short for ResNet.

| Settings | Heterogeneous Feature Extractors | | | | Heterogeneous Classifiers | | Large Client Amount | |
|---|---|---|---|---|---|---|---|---|
| | $MHMG_2$ | $MHMG_3$ | $MHMG_4$ | $MHMG_9$ | Res34-HtC4 | $MHMG_8$-HtC4 | 50Clients | 100Clients |
| FedGen | 44.33±0.07 | 43.52±0.45 | 42.76±0.17 | 42.25±0.04 | - | - | 37.80±0.17 | 36.66±0.02 |
| FedKD | 45.24±0.67 | 46.00±0.55 | 44.15±0.42 | 40.92±1.55 | 37.78±0.49 | 38.21±1.13 | 34.65±1.33 | 32.01±0.10 |
| FedProto | 40.04±0.14 | 36.50±0.21 | 38.26±0.15 | 27.02±0.19 | 34.42±0.14 | 30.27±0.12 | 18.90±0.55 | 14.45±0.09 |
| FedGH | 44.09±0.10 | 43.02±0.13 | 42.74±0.20 | 42.05±0.18 | 38.45±0.06 | 37.61±0.14 | 37.39±0.33 | 36.55±0.17 |
| FedTGP | 46.45±0.18 | 45.03±0.25 | 43.46±0.44 | 38.45±0.09 | 37.22±0.27 | 38.39±0.24 | 36.12±0.23 | 34.72±0.05 |
| FedORGP | 45.73±0.17 | 44.13±0.32 | 44.21±0.29 | 41.97±0.33 | 40.62±0.46 | 39.91±0.35 | 38.10±0.18 | 36.67±0.29 |
| **Ours** | **50.86±0.26** | **50.32±0.07** | **48.35±0.21** | **43.20±0.26** | **49.51±0.10** | **47.12±0.11** | **41.89±0.37** | **37.93±0.44** |

### 4.3 ROBUSTNESS TO MODEL HETEROGENEITY

Table 2 evaluates the robustness of our method on CIFAR-100 under practical settings with different types of model heterogeneity. For heterogeneous feature extractors, our method achieves 50.86%, 50.32%, 48.35%, and 43.20% accuracy in $MHMG_2$, $MHMG_3$, $MHMG_4$, and $MHMG_9$, respectively. Compared with Fed-Proto, our method shows much smaller performance degradation (7.66% v.s. 17.53%) as the number of architectures increases, indicating stronger resilience to diverse client models.

We further consider classifier heterogeneity. In the Res34-$HtC_4$ setting, our method reaches 49.51% accuracy, clearly outperforming all baselines. More importantly, in the challenging $MHMG_8$-$HtC_4$ setting, which combines eight heterogeneous feature extractors with four heterogeneous classifiers, FedDOR achieves 47.12%, outperforming FedProto by 16.85%. These results confirm our method's effectiveness in handling dual heterogeneity, a common challenge in real-world FL deployments.

### 4.4 IMPACT OF LARGE CLIENTS

Under large-scale settings with partial participation ($\rho = 0.5$), our method demonstrates excellent scalability and stability. With 50 clients, we achieve 41.89% accuracy, surpassing FedORGP by 3.79% and FedTGP by 5.77%. As the system scales to 100 clients, our method maintains 37.93% accuracy, showing only a 3.96% performance decrease compared to the 50-client setting. This represents the smallest performance degradation among all compared methods, highlighting our approach's particular suitability for real-world cross-device federated learning applications where partial participation and large client populations are the norm rather than the exception. Furthermore, the consistent margins over baselines suggest that our method effectively mitigates the adverse effects of client drift and stochastic participation. These findings underscore its robustness in highly dynamic federated environments.

### 4.5 ROBUSTNESS TO NON-IID DATA

To evaluate the robustness of our method under varying degrees of statistical heterogeneity among clients. As shown in Table 3, our method consistently achieves the best accuracy across all non-IID levels. In the most challenging case ($\alpha = 0.01$), it still reaches 72.17%, while competing methods fall notably behind. Moreover, even under moderate heterogeneity (e.g., $\alpha = 0.3$), our method outperforms the best baseline by more than 5%, indicating stable improvements across different scenarios. These results demonstrate that our orthogonality-constrained prototypes not only preserve class separability under severe distribution shifts but also enable more reliable knowledge transfer across clients.

Table 3: Test accuracy (%) on Cifar100 in the practical setting using the MHMG$_8$ model group with data distributions under various degrees of non-IID ($\alpha$) and feature dimensions ($K$).

| Settings | Data Distribution Non-IID Degrees | | | | Feature Dimension | | | |
|---|---|---|---|---|---|---|---|---|
| | $\alpha$ = 0.5 | $\alpha$ = 0.3 | $\alpha$ = 0.1 | $\alpha$ = 0.01 | $K$ = 128 | $K$ = 256 | $K$ = 512 | $K$ = 1024 |
| FedGen | 21.92±0.10 | 27.76±0.17 | 40.14±0.33 | 65.89±0.07 | 39.23±0.14 | 39.51±0.16 | 40.14±0.33 | 40.34±0.17 |
| FedKD | 21.33±0.91 | 27.45±0.77 | 39.25±1.43 | 63.78±2.01 | 38.79±1.26 | 39.60±1.47 | 39.25±1.43 | 39.32±1.47 |
| FedProto | 17.44±0.26 | 21.95±0.26 | 34.37±0.18 | 56.84±0.35 | 30.25±0.07 | 33.16±0.08 | 34.37±0.18 | 33.88±0.23 |
| FedGH | 21.72±0.17 | 27.62±0.35 | 39.74±0.22 | 65.70±0.22 | 38.99±0.17 | 39.36±0.10 | 39.74±0.22 | 39.85±0.43 |
| FedTGP | 19.95±0.38 | 26.14±0.25 | 38.99±0.27 | 66.42±0.22 | 36.99±0.18 | 37.97±0.20 | 38.99±0.27 | 39.25±0.30 |
| FedORGP | 21.76±0.09 | 27.27±0.26 | 40.71±0.18 | 68.45±0.32 | 39.90±0.19 | 40.94±0.39 | 40.71±0.18 | 40.63±0.14 |
| **Ours** | **25.89±0.24** | **32.86±0.14** | **47.06±0.27** | **72.17±0.27** | **45.43±0.48** | **46.52±0.44** | **47.06±0.27** | **47.00±0.14** |

## 4.6 IMPACT OF FEATURE DIMENSION

To evaluate the impact of feature space capacity on model performance, we conduct experiments with different feature dimensions ($K = 128$ to $K = 1024$) on Cifar100 using the MHMG$_8$ model group. As a result, as shown in Table 3, our method consistently outperforms six baseline approaches across all dimensions. Even under the most constrained setting ($K = 128$), it already surpasses FedORGP and FedTGP by a clear margin. These results indicate that our method can efficiently learn discriminative representations even in limited feature spaces, maintaining stable performance as feature dimensions increase.

## 4.7 ABLATION STUDY

Table 4: The test accuracy (%) in the practical setting using the MHMG$_8$ model group for ablation study.

| | FedProto | w/o OI | w/o DOR | **Ours** |
|---|---|---|---|---|
| Cifar100 | 34.37±0.18 | 40.71±0.18 | 38.64±1.00 | **47.06±0.27** |
| Flowers102 | 23.14±0.66 | 48.74±0.22 | 49.49±0.73 | **52.39±0.71** |
| Tiny-Imagenet | 15.10±0.40 | 21.84±0.17 | 22.01±0.57 | **28.63±0.19** |

Table 4 reports the ablation study on three datasets, verifying the necessity of orthogonal initialization (OI) and dual orthogonal regularization (DOR). Removing either component leads to substantial accuracy drops, with DOR having the stronger impact on prototype separability.Our complete method consistently achieves the best results across all datasets. The gains are more pronounced on complex datasets such as Flowers102 and Tiny-Imagenet, indicating that orthogonal constraints are particularly effective when semantic structures are diverse. These findings confirm that OI and DOR complement each other: OI provides stable initialization, while DOR enforces discriminative global prototypes, together enhancing robustness to client heterogeneity.

## 5 CONCLUSION

In this work, we propose FedDOR, a novel HFL method that uses orthogonal initialization and dual orthogonality regularization to learn maximally discriminative global prototypes. By enforcing both initial and ongoing geometric constraints, FedDOR achieves stronger inter-class separation and semantic consistency than prior prototype-based methods. Experiments show that FedDOR consistently outperforms all baselines across heterogeneous settings, demonstrating superior accuracy, robustness, and communication efficiency.

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

## A APPENDIX

### A.1 THE USE OF LARGE LANGUAGE MODELS

The language of this paper was polished using large language models (LLMs) to enhance clarity and readability. The final content and academic integrity remain the responsibility of the authors.

## A.2 HETEROGENEOUS FEDERATED LEARNING

The landscape of Heterogeneous Federated Learning encompasses multiple methodological families, each with unique characteristics and limitations. Early systematic approaches addressed hardware heterogeneity through shared architectural designs. Among these, LG-FedAvg (McMahan et al., 2017) proposed keeping local lower layers private while aggregating only higher-level layers across participating clients. This method reduced computational overhead but still required maintaining a consistent architectural framework across all devices, which presented practical limitations in fully heterogeneous environments and raised potential privacy concerns regarding model structure exposure.

Knowledge distillation methods emerged as a promising alternative to overcome architectural constraints. FedMD (Li & Wang, 2019) pioneered this approach by utilizing a public dataset to align model predictions across heterogeneous clients through logit matching. FedDF (Lin et al., 2020) extended this concept by employing ensemble distillation on the server side to improve model generalization. While effective under ideal conditions, these methods face significant practical limitations due to their dependence on public datasets, which are often unavailable in real-world federated learning scenarios due to privacy regulations and data accessibility constraints. The quality and representativeness of these public datasets substantially influence the performance, making these methods unsuitable for many practical applications.

The evolution of HFL methods continues to address these challenges through innovative approaches that balance performance, privacy, and practical applicability. Current research trends indicate growing interest in personalized federated learning, self-supervised techniques, and more sophisticated aggregation mechanisms that can better handle the complex interplay between statistical and system heterogeneity in real-world deployment scenarios.

## A.3 PROTOTYPE LEARNING.

Prototype learning has been widely adopted in both centralized and federated settings to learn representative embeddings for each class. In conventional centralized learning, prototypes help improve the discriminability of features through metric learning constraints (Dai et al., 2023; Snell et al., 2017; Michieli & Ozay, 2021). In federated learning, prototypes serve as efficient carriers of knowledge that can be aggregated across clients without sharing raw data or model parameters(Tan et al., 2022b). FedProto (Tan et al., 2022b) proposed to compute local prototypes by averaging features of each class and aggregate them on the server via weighted averaging. Nonetheless, naive aggregation often fails to preserve semantic relationships, especially under highly non-IID data. FedTGP (Zhang et al., 2024b) improved prototype discrimination using adaptive-margin-enhanced contrastive learning, which leverages contrastive loss to maximize inter-prototype Euclidean margins(Hayat et al., 2019). Still, these methods do not fully exploit geometric properties such as orthogonality to ensure inter-class separation and semantic consistency.

## A.4 ORTHOGONAL PROTOTYPE GENERATION.

As an effective inductive deviation, orthogonality is widely used in machine learning models to reduce feature redundancy and enhance discriminant ability (Qiu et al., 2023). Under the setting of centralized deep learning, it has been proved that applying orthogonal constraints to the weight matrix or feature representation can improve training stability and generalization performance
(Shang et al., 2024). Recent studies, such as FedSOL (Lee et al., 2024) and FedORGP (Guo et al., 2025), further introduced the loss function based on orthogonality to promote greater angular boundary between categories. However, these methods are obviously limited in the environment of orthogonal prototype generation. Its representation space lacks clear structural prior and is sensitive to parameter initialization, especially in highly heterogeneous scenes. In our FedDOR, the server integrates the prototype-like feature generation and orthogonal constraint mechanism, and innovatively adopts orthogonal initialization strategy

to embed the prototype in the client, thus introducing the maximum separation degree and uniformly distributed geometric priors into the feature space, effectively promoting clear semantic separation.

## A.5 CHALLENGES IN PROTOTYPE AGGREGATION FOR HETEROGENEOUS FEDERATED LEARNING

However, naive aggregation mechanisms, such as weighted averaging, compromise the semantic integrity of prototypes (Zhang et al., 2024b; Dai et al., 2023), often causing representation collapse and inter-class overlap in the feature space, especially under high heterogeneity (Tan et al., 2022b; Dai et al., 2023). While recent advances like FedTGP (Zhang et al., 2024b) have employed contrastive learning to improve prototype discrimination through Euclidean margin maximization, such approaches remain limited in preserving semantic structure and are misaligned with the angular characteristics of standard classification losses. As illustrated in Fig. 1(a), although FedTGP enhances distinctiveness by increasing inter-class prototype margins, it still fails to achieve sufficient separation between certain categories in the feature space. Another orthogonal strategy is introduced in FedORGP (Guo et al., 2025), which applies orthogonality regularization via a loss function to encourage angular separation among global prototypes on the server side. Although this method improves inter-class discriminability to some extent, it relies solely on soft constraints during training and does not enforce a geometric prior in the representation space. As shown in Fig. 1(b), FedORGP still exhibits feature aliasing between some classes, indicating limited generalization under distribution shifts. As a result, FedORGP remains sensitive to initialization and may converge to suboptimal prototype configurations, especially under highly heterogeneous settings.

In order to solve the limitations of existing prototype-based heterogeneous federated learning (HtFL) methods, we propose a new method FedDOR based on class prototype feature generation. This method integrates class prototype generation module and Dual Orthogonal Regularization (DOR) function on the server side to ensure global semantic integrity. On the client side, we innovatively adopt orthogonal initialization prototype embedding, and provide consistent and more discriminating optimization objectives for the local model by imposing maximum separation and uniformly distributed prior geometric constraints in the feature space. Specifically, this method projects features and prototypes onto the unit hypersphere by L2 normalization, and initializes the classifier by orthogonal weights, which explicitly promotes the feature separation between classes. At the same time, the learnable scaling factor is introduced to further optimize the decision boundary. In addition, intra-class orthogonality loss enhances intra-class compactness by aligning samples with their prototypes, while inter-class orthogonality loss realizes explicit inter-class separation by suppressing the similarity with mismatched prototypes. With the cooperation of the above components, FedDOR effectively expands the inter-class interval of client features and overcomes the inherent problems such as initialization sensitivity and category overlap. As shown in Figure 1(c), the original indistinguishable categories under FedDOR framework are effectively separated by significantly improved decision margins and clear structural consistency, thus showing superior and stable convergence performance in a highly heterogeneous federal environment.

## A.6 LIMITATIONS OF NAIVE PROTOTYPE AGGREGATION

We consider a typical Heterogeneous Federated Learning (HFL) system comprising a central server and $m$ clients, where each client $k$ holds a private dataset $\mathcal{D}_k$ and employs a model with heterogeneous architecture. For the client $k$, the local model is typically divided into a feature extractor $f_k(\cdot; \phi_k)$ parameterized by $\phi_k$ and a classifier $h_k(\cdot; \theta_k)$ parameterized by $\theta_k$. The objective of HFL is to collaboratively train these heterogeneous local models without exchanging private data or model parameters, while mitigating the performance degradation caused by both statistical and model heterogeneity.

Following the prototype-based HFL paradigm (Tan et al., 2022b), clients compute and upload class prototypes to the server as a form of knowledge exchange. The prototype for class $c$ on client $k$ is calculated as

the mean of feature representations:

$$\mathbf{p}_k^c = \frac{1}{|\mathcal{D}_k^c|} \sum_{(x_i, y_i) \in \mathcal{D}_k^c} f_k(x_i; \phi_k), \tag{5}$$

where $\mathcal{D}_k^c$ denotes the subset of data samples belonging to class $c$ on client $k$.

On the server side, the global prototype of class $c$ is commonly obtained by weighted averaging as $\bar{\mathbf{p}}^c = \sum_{k=1}^m \frac{|\mathcal{D}_k^c|}{N^c} \mathbf{p}_k^c$, where $N^c$ is the total number of samples of class $c$ across all clients. However, this naive aggregation suffers from two critical drawbacks. First, it requires clients to upload their private class distribution information ($|\mathcal{D}_k^c|$), raising privacy concerns. Second, averaging prototypes across heterogeneous models inevitably produces inconsistent magnitudes and directions, which result in "prototype collapse". Consequently, the global prototypes exhibit diminished inter-class separation and weakened semantic fidelity, ultimately providing suboptimal guidance for local training and degrading overall performance.

Motivated by these limitations, we propose a novel method that fundamentally rethinks the initialization and learning of prototypes, ensuring that the global prototypes remain maximally separable and semantically meaningful during the federated training process.

### A.7 LOCAL MODEL UPDATE WITH GLOBAL PROTOTYPE GUIDANCE

To encourage the learned feature representations $f_k(x; \phi_k)$ to align tightly with the global prototype $\bar{\mathbf{p}}^y$ of its original annotation signal $y$, we further construct the prototype alignment loss $\mathcal{L}_{\text{align}}$ to optimize inter-class separation via cosine similarity:

$$\mathcal{L}_{\text{align}} = \frac{1}{2B} \sum_{i=1}^B \left\| \frac{f_k(x_i; \phi_k)}{\|f_k(x_i; \phi_k)\|_2} - \frac{\bar{\mathbf{p}}^{y_i}}{\|\bar{\mathbf{p}}^{y_i}\|_2} \right\|_2^2, \tag{6}$$

where $B$ means the training batch size. Based on Eq. 6, we effectively minimize the Euclidean distance between the normalized feature vectors and their corresponding normalized prototypes. Mathematically, such an objective is equivalent to maximizing cosine similarity, offering an alternative perspective on alignment.

During local training, each client employs stochastic gradient descent to optimize its model parameters as $\theta^{(t+1)} = \theta^{(t)} - \eta \nabla_\theta \mathcal{L}_{\text{local}}$ where $\eta$ represents the learning rate that controls the step size of the local optimization. After local training, each prototype $\mathbf{p}_k^c \in \mathbb{R}^d$ is calculated by the mean vector of the features belonging to the same class. These updated prototypes are then sent to the server for aggregation in the next communication round.

This local update mechanism ensures that each client's model not only fits its local data but also remains consistent with the global semantic structure represented by the prototypes, thereby improving both personalization and generalization in the federated learning process.

### A.8 TRAINING CONFIGURATION

All experiments are carried out on a Linux workstation based on the x86_64 architecture with Ubuntu as the operating system. The machine is equipped with an NVIDIA GeForce RTX 3090 GPU (24 GB VRAM) paired with CUDA 12.1, alongside 64 GB of system memory. This hardware configuration provides sufficient computational power and memory bandwidth to support large-scale federated learning tasks.

In our experimental setup, unless otherwise mentioned, we simulate 20 clients with a participation ratio of $\rho = 1$. Both the client-side models and the central server adopt the SGD optimizer with a fixed learning rate

of 0.01. Local updates are performed for a single epoch in each communication round, with a batch size of 32 applied to both client training ($B$) and server aggregation ($B_p$). By default, we employ the MHMG$_8$ model group to introduce architectural diversity.

To study statistical heterogeneity, two data partitioning strategies are considered. In the pathological case, clients are assigned highly unbalanced label distributions (2/10/10/20 classes per client) drawn from Cifar-10, Cifar-100, Flowers102, and Tiny-ImageNet. In the practical case, client data are sampled according to a Dirichlet distribution with concentration parameter $\alpha = 0.1$, which better reflects non-IID conditions in the real world. Each client dataset is split into 75% for training and 25% for testing. For each algorithm, we repeat the training three times, each run lasting 100 communication rounds, and report the mean and variance of the highest test accuracy achieved.

Table 5: Test accuracy (%) on Cifar100 in the practical setting using the MHMG$_8$ model group with different client training epochs ($E$).

| Settings | Client Training Epochs | | | |
|---|---|---|---|---|
| | $E = 1$ | $E = 5$ | $E = 10$ | $E = 20$ |
| FedGen | 40.14±0.33 | 40.27±0.22 | 40.43±0.23 | 40.87±0.13 |
| FedKD | 39.25±1.43 | 41.05±0.19 | 40.21±0.14 | 39.11±0.22 |
| FedProto | 34.37±0.18 | 37.57±0.42 | 37.91±0.09 | 37.25±0.46 |
| FedGH | 39.74±0.22 | 40.21±0.06 | 40.56±0.07 | 40.80±0.16 |
| FedTGP | 38.99±0.27 | 41.09±0.17 | 41.98±0.44 | 45.23±0.22 |
| FedORGP | 40.71±0.18 | 41.28±0.18 | 41.60±0.53 | 41.69±0.38 |
| **Ours** | **47.06±0.27** | **47.11±0.27** | **46.99±0.10** | **47.19±0.31** |

## A.9  IMPACT OF CLIENT TRAINING EPOCHS

To evaluate the impact of local computation on model performance, we analyze the effect of varying numbers of client training epochs ($E$) using the Cifar100 dataset under the practical setting with the MHMG$_8$ model group. s shown in the part of client training rounds in Table 5, we compare our method against six federated learning approaches with $E$ ranging from 1 to 20, measuring their stability and robustness under extended local training.

Our method demonstrates exceptional stability across all epoch settings, maintaining consistently high accuracy with minimal fluctuation. We achieve 47.06%, 47.11%, 46.99%, and 47.19% accuracy at $E = 1$, 5, 10, and 20 respectively, indicating remarkable resistance to client drift or overfitting despite increased local computation. In contrast, other methods exhibit varying degrees of performance instability: FedKD shows significant fluctuation between 39.25% and 41.05% accuracy, while FedTGP, despite showing consistent improvement from 38.99% to 45.23% with more epochs, still trails our method by 1.96% even at its peak performance ($E = 20$). FedProto also demonstrates sensitivity to epoch settings, with its accuracy varying between 34.37% and 37.91%.

## A.10  HYPERPARAMETER SENSITIVITY

The purpose of this experiment is to examine how different weightings of the classification loss $\lambda_{CE}$ and the alignment loss $\lambda_{align}$ affect the overall performance of the model. As shown in Fig. 3, the results, averaged over three independent runs, indicate that maintaining a moderate balance between the two loss components consistently leads to better accuracy across all four datasets. The configuration with $\lambda_{CE} = 10$

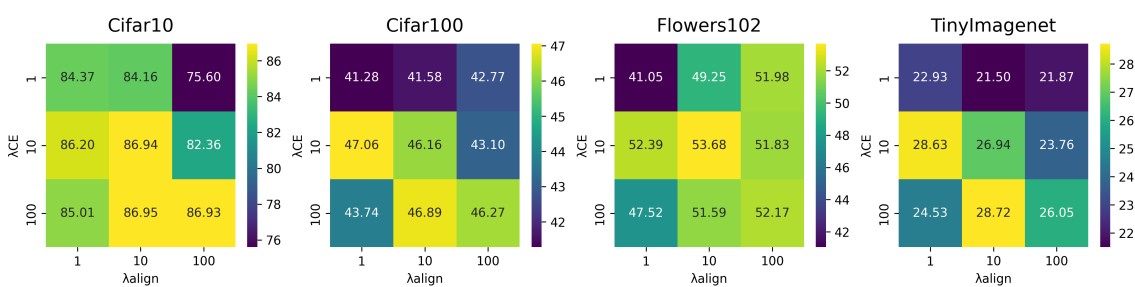

Figure 3: The test accuracy (%) across different datasets under the practical setting, using the MHMG8 model group with varying $\lambda_{CE}$ or $\lambda_{align}$.

and $\lambda_{align}$ = 1 achieves strong results, such as 86.20% on Cifar10 and 47.06% on Cifar100. In contrast, assigning excessively large weights to either loss term, for example $\lambda_{align}$ = 100 or $\lambda_{CE}$ = 100, generally reduces accuracy, suggesting that overemphasizing one objective disrupts the balance between discriminative learning and feature alignment. These observations highlight the importance of choosing well balanced hyperparameters to ensure stable and effective model training.

## A.11 IMPACT OF EXTREME PATHOLOGICAL ENVIRONMENT

Table 6: Test accuracy (%) on Cifar100 in the extreme pathological environment with different numbers of classes per client.

| Settings | Number of Classes per Client | | | |
|---|---|---|---|---|
| | 5 | 10 | 15 | 20 |
| FedGen | $71.49 \pm 0.20$ | $55.19 \pm 0.20$ | $46.09 \pm 0.28$ | $37.75 \pm 0.30$ |
| FedKD | $69.17 \pm 1.43$ | $52.42 \pm 1.24$ | $44.90 \pm 0.83$ | $33.90 \pm 1.07$ |
| FedProto | $70.28 \pm 0.19$ | $52.62 \pm 0.11$ | $43.75 \pm 0.10$ | $35.46 \pm 0.36$ |
| FedGH | $71.48 \pm 0.15$ | $54.81 \pm 0.12$ | $46.17 \pm 0.22$ | $37.66 \pm 0.27$ |
| FedTGP | $69.83 \pm 0.20$ | $53.37 \pm 0.31$ | $45.04 \pm 0.35$ | $35.09 \pm 0.29$ |
| FedORGP | $74.84 \pm 0.33$ | $58.29 \pm 0.10$ | $48.43 \pm 0.56$ | $39.50 \pm 0.23$ |
| **Ours** | $\mathbf{77.35 \pm 0.26}$ | $\mathbf{63.21 \pm 0.28}$ | $\mathbf{53.56 \pm 0.40}$ | $\mathbf{46.48 \pm 0.28}$ |

To assess the robustness of different federated learning methods under extreme data heterogeneity, we evaluate their performance on CIFAR-100 with varying numbers of classes per client, as shown in Table 6. Specifically, the number of classes per client ranges from 5 to 20, representing increasingly balanced data distributions. As expected, all methods experience performance degradation as the data becomes more skewed. However, FedDOR consistently achieves the best accuracy across all settings, significantly outperforming six strong baselines. Notably, even in the most pathological case (5 classes per client), FedDOR maintains a clear performance margin, demonstrating its superior capability to learn robust and generalizable representations under highly non-IID conditions.

Table 7: Test accuracy (%) on Cifar100 in the practical setting using the MHMG$_8$ model group with different number of clients (M) and join ratio ($\rho$).

| Settings | Number of Clients (M) and Join Ratio ($\rho$) | | | |
|---|---|---|---|---|
| | $M = 40, \rho = 50\%$ | $M = 50, \rho = 40\%$ | $M = 80, \rho = 25\%$ | $M = 100, \rho = 20\%$ |
| FedGen | $40.16 \pm 0.16$ | $37.87 \pm 0.05$ | $34.41 \pm 0.43$ | $35.18 \pm 0.02$ |
| FedKD | $37.06 \pm 1.30$ | $34.05 \pm 0.73$ | $32.84 \pm 1.30$ | $31.25 \pm 0.04$ |
| FedProto | $21.55 \pm 0.13$ | $15.06 \pm 0.21$ | $13.90 \pm 0.84$ | $13.16 \pm 0.55$ |
| FedGH | $39.67 \pm 0.25$ | $37.38 \pm 0.30$ | $34.12 \pm 0.23$ | — |
| FedTGP | $38.38 \pm 0.32$ | $36.01 \pm 0.02$ | $33.06 \pm 0.09$ | $34.19 \pm 0.14$ |
| FedORGP | $40.98 \pm 0.16$ | $38.08 \pm 0.20$ | $35.08 \pm 0.08$ | $36.49 \pm 0.49$ |
| **Ours(32)** | $\mathbf{45.46 \pm 0.34}$ | $\mathbf{41.09 \pm 0.60}$ | $\mathbf{38.90 \pm 0.13}$ | $\mathbf{38.86 \pm 0.12}$ |

## A.12 IMPACT OF CLIENT SCALE AND PARTICIPATION RATIO

To evaluate the scalability and robustness of different federated learning methods under practical deployment conditions, we conduct experiments on CIFAR-100 using the MHMG$_8$ model group for 200 communication rounds, with varying numbers of clients ($M$) and client join ratios ($\rho$), as shown in Table 7. As the total number of clients increases and the participation ratio decreases, the overall performance of all methods tends to decline, reflecting the increasing difficulty in maintaining global model consistency with limited client participation. It is noteworthy that FedGH fails to converge under the most challenging configuration ($M = 100$, $\rho = 20\%$), indicating its instability in large-scale and sparsely participating environments. In contrast, FedDOR consistently outperforms all baselines across all settings and maintains a stable accuracy even in this extreme case, demonstrating strong scalability and resilience to partial participation in large-scale federated systems.

## A.13 EFFECT ON SMALL-SCALE DATASETS

Table 8: Test accuracy (%) on MNIST and Fashion-MNIST under pathological setting and practical setting using the MHMG$_8$ model group.

| Settings | Pathological Setting | | Practical Setting | |
|---|---|---|---|---|
| | MNIST | Fashion-MNIST | MNIST | Fashion-MNIST |
| FedGen | $99.44 \pm 0.03$ | $99.21 \pm 0.02$ | $98.55 \pm 0.04$ | $96.55 \pm 0.06$ |
| FedKD | $98.99 \pm 0.29$ | $98.92 \pm 0.23$ | $97.90 \pm 0.47$ | $96.45 \pm 0.11$ |
| FedProto | $99.33 \pm 0.04$ | $99.15 \pm 0.01$ | $\mathbf{99.34 \pm 0.04}$ | $86.85 \pm 0.28$ |
| FedGH | $99.46 \pm 0.02$ | $99.19 \pm 0.05$ | $98.59 \pm 0.03$ | $96.44 \pm 0.01$ |
| FedTGP | $\mathbf{99.70 \pm 0.02}$ | $\mathbf{99.31 \pm 0.02}$ | $99.22 \pm 0.05$ | $\mathbf{96.78 \pm 0.07}$ |
| FedORGP | $99.56 \pm 0.05$ | $99.30 \pm 0.06$ | $98.62 \pm 0.08$ | $96.64 \pm 0.09$ |
| **Ours** | $99.43 \pm 0.01$ | $99.05 \pm 0.08$ | $98.25 \pm 0.10$ | $96.01 \pm 0.18$ |

To evaluate the stability of the methods on small-scale datasets, we conduct additional evaluations on two small-scale datasets, MNIST and Fashion-MNIST, under both pathological and practical settings, as shown in Table 8. Overall, all methods achieve high accuracy on these datasets, reflecting the relative simplicity of the tasks. Our method, FedDOR, performs competitively with other baselines, and the performance

differences among methods are minimal. These results indicate that while FedDOR demonstrates clear advantages on large-scale datasets, it also maintains reliable and comparable performance on smaller-scale tasks.

## A.14 LIMITATIONS

While FedDOR demonstrates robust performance across various heterogeneous federated learning scenarios, we acknowledge specific limitations regarding the feature space capacity. Specifically, when the feature dimension $d$ is significantly smaller than the number of categories $C$ (i.e., $d < C$), strictly orthogonal prototypes cannot be constructed in the Euclidean space $\mathbb{R}^d$. In such "bottleneck" scenarios, the orthogonality constraints may lead to optimization difficulties or suboptimal equiangularity rather than true orthogonality, which limits the expressiveness of the learned representations. Future work may explore adaptive dimension expansion strategies or relax the strict orthogonality to quasi-orthogonality (e.g., via Equiangular Tight Frames bounds) to mitigate this limitation in highly constrained resource environments.

## A.15 LOCAL STABILITY OF ORTHOGONAL INITIALIZATION

### A.15.1 PRELIMINARIES AND NOTATION

We adhere to the notation from the main paper. Key elements are summarized and supplemented as follows:

- Classification Setup: $C$ is the number of classes. $d$ is the feature dimension.

- Prototypes: $W = \{w_c\}_{c=1}^{C} \in \mathbb{R}^{C \times d}$ is the matrix of trainable prototype embeddings. Following the FedDOR algorithm, all prototypes $w_c$ and feature vectors $f(x)$ are L2-normalized onto the unit sphere $\mathbb{S}^{d-1}$. We denote normalized vectors as $\hat{w}_c = w_c / \|w_c\|$.

- Features and Classifier: $R = \{r_j\}$ denotes a set of (normalized) feature vectors from a representative data batch. $\Theta$ collectively represents the parameters of the client-side classifiers.

- Loss Components: The total training objective combines:

  - Frame Potential (FP): $FP(W) = \sum_{i \neq j} \langle \hat{w}_i, \hat{w}_j \rangle^2$, which encourages an equiangular prototype configuration.

  - Alignment Loss: $\mathcal{L}_{\text{align}} = \sum_j \left\| \frac{r_j}{\|r_j\|} - \hat{w}_{y_j} \right\|^2$, which pulls features towards their assigned prototypes.

  - Cross-Entropy Loss: $\mathcal{L}_{\text{CE}}(R, \Theta)$ for classification.

- Total Energy Function: We analyze the dynamics of the following Lyapunov-like function:

$$E(W, R, \Theta) = FP(W) + \alpha \mathcal{L}_{\text{align}}(W, R) + \beta \mathcal{L}_{\text{CE}}(R, \Theta), \quad \alpha, \beta > 0.$$

- Equilibrium Point: $(W^*, R^*, \Theta^*)$ denotes the ideal equilibrium, where:

  1. $W^*$ is the OI configuration, approximating an equiangular tight frame (ETF).
  2. $R^*$ is perfectly aligned, meaning for any sample $j$ from class $c$, $r_j^* = \hat{w}_c^*$.
  3. $\Theta^*$ is an optimal classifier for the aligned features $R^*$.

- Tangent Space: $T_{\hat{w}_c} \mathbb{S}^{d-1}$ denotes the tangent space of the unit sphere at $\hat{w}_c$. Gradient flows for prototypes are considered as projections onto this manifold.

### A.15.2 ASSUMPTIONS

The analysis rests on the following standard assumptions:

- (A1) Smoothness: The function $E(W, R, \Theta)$ is twice continuously differentiable in a neighborhood of the equilibrium $(W^*, R^*, \Theta^*)$. Its gradients are Lipschitz continuous.

- (A2) Manifold Constraints: The dynamics respect the spherical constraints. Gradients w.r.t. $w_c$ are projected onto the tangent space $T_{\hat{w}_c}\mathbb{S}^{d-1}$.

- (A3) Stochastic Gradient Descent (SGD): The discrete SGD updates are unbiased estimators of the full gradient with bounded variance. The learning rate $\eta$ is sufficiently small to ensure the dynamics approximate the continuous gradient flow.

- (A4) OI as Local FP Minimizer: The orthogonal initialization $W^*$ is a strict local minimizer of the Frame Potential $FP(W)$ on the product of spheres $\prod_{c=1}^{C}\mathbb{S}^{d-1}$. This is justified by the construction of $W^*$ as an approximate equiangular tight frame (Benedetto & Fickus, 2003).

### A.15.3 SUPPORTING LEMMAS

**Lemma 1 (Strict Local Minimality of FP at OI).**

Under Assumption (A4), the Hessian of $FP(W)$ at $W^*$, when projected onto the product tangent space $\prod_{c=1}^{C} T_{\hat{w}_c^*}\mathbb{S}^{d-1}$, is positive definite. Consequently, $W^*$ is a strict local minimizer of $FP(W)$ on the manifold.

The Frame Potential is a smooth function on the compact product manifold $\prod_{c=1}^{C}\mathbb{S}^{d-1}$. For an exact equiangular tight frame, it is known that the Hessian is positive definite in directions orthogonal to the global isometry group (Benedetto & Fickus, 2003; Waldron, 2018). Since $W^*$ is a close approximation of an ETF by construction (Assumption A4), and the property of positive definiteness is stable under small perturbations, the Hessian at $W^*$ remains positive definite on the tangent space.

**Lemma 2 (Local Convexity of Alignment Loss).**

For a fixed set of features $R$ where each class $c$ has at least $d$ linearly independent samples, the alignment loss $\mathcal{L}_{\text{align}}(W, R)$ is locally strongly convex in each prototype $\hat{w}_c$ on the tangent space $T_{\hat{w}_c}\mathbb{S}^{d-1}$ near the aligned state $R = W$.

For a fixed $r_j$, the function $\hat{w} \mapsto \|\hat{w} - r_j\|^2$ is linear, hence convex. Its restriction to the sphere is a geodesically convex function near the point $r_j$. Summing over multiple samples per class ensures that the aggregate Hessian of $\mathcal{L}_{\text{align}}$ on the tangent space has full rank, implying strong convexity.

**Lemma 3 (Feature Dynamics Contract Towards Alignment).**

Consider the combined dynamics from the alignment and cross-entropy losses: $\dot{r}_j = -\nabla_{r_j}(\alpha\mathcal{L}_{\text{align}} + \beta\mathcal{L}_{\text{CE}})$. In a neighborhood of the equilibrium $(W^*, R^*, \Theta^*)$, the component of $\dot{r}_j$ along the direction of $(\hat{w}_{y_j}^* - r_j)$ is strictly positive, driving $r_j$ towards $\hat{w}_{y_j}^*$.

The gradient of the alignment loss directly pulls $r_j$ towards $\hat{w}_{y_j}$: $-\nabla_{r_j}\mathcal{L}_{\text{align}} \propto (\hat{w}_{y_j} - r_j)$. The gradient of the cross-entropy loss, $-\nabla_{r_j}\mathcal{L}_{\text{CE}}$, encourages $r_j$ to align with the classifier weight for the correct class. At equilibrium, this classifier weight is aligned with $\hat{w}_{y_j}^*$. By smoothness (A1), in a neighborhood of equilibrium, the CE gradient has a positive projection onto $(\hat{w}_{y_j}^* - r_j)$. Thus, the combined dynamics are contracting.

### A.15.4 MAIN THEOREM AND PROOF

**Theorem 1 (Local Asymptotic Stability of OI).**

Let $(W^*, R^*, \Theta^*)$ be the OI-induced equilibrium defined in Section A.15.1. Under Assumptions (A1)-(A4), there exists a neighborhood $U$ of this equilibrium such that for any initial condition $(W(0), R(0), \Theta(0)) \in U$, the continuous-time gradient flow

$$\dot{W} = -\nabla_W E, \quad \dot{R} = -\nabla_R E, \quad \dot{\Theta} = -\nabla_\Theta E$$

converges asymptotically to $(W^*, R^*, \Theta^*)$ as $t \to \infty$. Furthermore, under Assumption (A3), the discrete-time SGD iterations remain within the basin of attraction with high probability and the expected value of $E$ decreases monotonically for sufficiently small learning rates.

*Proof:*

We proceed in four steps.

**Step 1:** $(W^*, R^*, \Theta^*)$ **is a Stationary Point.**

At equilibrium, the following conditions hold by definition:

1. Features are aligned: $r_j^* = \hat{w}_{y_j}^*$ for all $j$, making $\nabla_R \mathcal{L}_{\text{align}} = 0$.
2. The classifier is optimal: $\nabla_\Theta \mathcal{L}_{\text{CE}} = 0$.
3. The gradient of $FP(W)$ vanishes at $W^*$ (Lemma 1), and the alignment term $\nabla_W \mathcal{L}_{\text{align}}$ also vanishes due to perfect alignment.

Thus, $\nabla E(W^*, R^*, \Theta^*) = 0$, confirming it is a stationary point.

**Step 2: Positive Definiteness of the Restricted Hessian at Equilibrium.**

Consider the Hessian $\nabla^2 E(W^*, R^*, \Theta^*)$. We analyze its structure in the combined variable space $Z = (W, R, \Theta)$, projecting gradients for $W$ and $R$ onto their respective tangent spaces.

- $W$-Block ($\mathbf{H}_{WW}$): By Lemma 1, $\nabla_{\hat{W}}^2 FP(W^*)$ is positive definite on the tangent space. The Hessian of $\mathcal{L}_{\text{align}}$ is also positive semidefinite (Lemma 2). Thus, $\mathbf{H}_{WW} = \nabla_{\hat{W}}^2 FP + \alpha \nabla_{\hat{W}}^2 \mathcal{L}_{\text{align}}$ is positive definite.
- $R$-Block ($\mathbf{H}_{RR}$): The Hessian $\nabla_{\hat{R}}^2 (\alpha \mathcal{L}_{\text{align}} + \beta \mathcal{L}_{\text{CE}})$ is positive definite due to the strong convexity of $\mathcal{L}_{\text{align}}$ (Lemma 2) and the positive contribution from $\mathcal{L}_{\text{CE}}$ near the optimum.
- Cross-Blocks ($\mathbf{H}_{WR}, \mathbf{H}_{RW}$, etc.): These blocks are bounded due to smoothness (A1).

Let $\mathbf{H} = \begin{bmatrix} \mathbf{H}_{WW} & \mathbf{H}_{WR} & \mathbf{H}_{W\Theta} \\ \mathbf{H}_{RW} & \mathbf{H}_{RR} & \mathbf{H}_{R\Theta} \\ \mathbf{H}_{\Theta W} & \mathbf{H}_{\Theta R} & \mathbf{H}_{\Theta\Theta} \end{bmatrix}$ be the full Hessian. Since the diagonal blocks $\mathbf{H}_{WW}$ and $\mathbf{H}_{RR}$ are positive definite, and the cross-terms are bounded, there exists a constant $\gamma > 0$ such that if the norms of the cross-term blocks are sufficiently small relative to the minimal eigenvalues of the diagonal blocks, then $\mathbf{H} + \mathbf{H}^\top$ is positive definite. This condition can be ensured by choosing the weights $\alpha$ and $\beta$ appropriately or by the structure of the problem near equilibrium. Consequently, the Jacobian of the gradient flow, $-\mathbf{H}$, is negative definite.

**Step 3: Local Asymptotic Stability of the Nonlinear System.**

The gradient flow $\dot{Z} = -\nabla E(Z)$ is a $C^1$ vector field. Since its linearization at $Z^*$ is $\delta \dot{Z} = -\mathbf{H} \delta Z$ and $-\mathbf{H}$ is a negative definite matrix (from Step 2), all eigenvalues have negative real parts. By Lyapunov's indirect

method (Khalil & Grizzle, 2002), the equilibrium $Z^*$ is locally asymptotically stable for the nonlinear system. This means there exists a neighborhood $U$ such that any trajectory starting in $U$ converges to $Z^*$ as $t \to \infty$.

**Step 4: Stability under Discrete SGD.**

Under Assumption (A3), the discrete SGD updates form a stochastic approximation of the gradient flow. Standard results in stochastic approximation (Kushner & Yin, 2003) state that if the equilibrium is locally asymptotically stable for the flow, the noise is zero-mean with bounded variance, and the learning rate $\eta$ is sufficiently small (or follows a Robbins-Monro schedule), then the iterates will converge to the equilibrium with high probability. More precisely, for small $\eta$, the expected energy satisfies $\mathbb{E}[E_{t+1}] \leq \mathbb{E}[E_t] - c\eta\mathbb{E}[\|\nabla E\|^2] + O(\eta^2)$, ensuring a decrease in expectation and confinement to the basin of attraction.

### A.16 ANALYSIS OF ADDITIONAL COMPUTATIONAL COST

This section provides a quantitative analysis of the computational overhead introduced by the two core components of FedDOR: the one-time client-side Orthogonal Initialization (OI) and the recurrent server-side prototype generation module $G(E; \Omega_G)$. The analysis confirms that these costs are negligible within a typical federated learning workload.

#### A.16.1 COMPUTATIONAL COST OF ORTHOGONAL INITIALIZATION (OI)

The Orthogonal Initialization is a one-time, pre-training step performed by each client. A standard and computationally efficient method to generate the $C \times d$ orthogonal prototype matrix involves generating a random matrix from a Gaussian distribution and applying QR decomposition.

- Complexity: The computational complexity of QR decomposition for a matrix of size $C \times d$ (assuming $d \geq C$) using Householder transformations is approximately $O(C^2 d)$ floating-point operations (FLOPs).

- Concrete Example: For a task like CIFAR-100, where the number of classes $C = 100$ and the feature dimension $d = 512$, the total FLOPs for OI are $100^2 \times 512 = 5.12 \times 10^6$ FLOPs ($\approx 5.12$ MFLOPs).

- Comparison: The cost of a single local training step for a ResNet-18 model (processing a batch of 32 images) is approximately $1.15 \times 10^{11}$ FLOPs ($\approx 115$ GFLOPs).

- Conclusion: The one-time OI cost ($\approx 5$ MFLOPs) is over four orders of magnitude smaller than the cost of processing a single batch during local training ($\approx 115,000$ MFLOPs). Its contribution to the total computational budget is therefore insignificant.

#### A.16.2 COMPUTATIONAL COST OF THE SERVER-SIDE MODULE $G(E; \Omega_G)$

The server-side module $G$ is a lightweight neural network designed to refine the global prototypes in each communication round. It consists of two fully connected layers with a ReLU activation, modeled as $d \to d_{\text{hidden}} \to d$, and processes each of the $C$ prototypes.

- **Complexity**: The FLOPs for a forward pass through two fully connected layers for $C$ vectors are approximately $C \times (2dd_{\text{hidden}} + 2d_{\text{hidden}}d) = 4Cdd_{\text{hidden}}$. Considering a backward pass cost factor of 3, the total cost for one update (forward + backward) is approximately $16Cdd_{\text{hidden}}$ FLOPs.

- **Concrete Example**: Using $C = 100$, $d = 512$, and a hidden dimension $d_{\text{hidden}} = 512$, the total FLOPs for $G$ per round are $16 \times 100 \times 512 \times 512 \approx 4.2 \times 10^8$ FLOPs ($\approx 420$ MFLOPs).

- **Comparison**: The total computation for a single client during one local epoch on CIFAR-100 (with $\approx 2000$ images) is approximately $7.2 \times 10^{12}$ FLOPs ($\approx 7.2$ TFLOPs).

- **Conclusion**: The server's computational work for module $G$ ($\approx 420$ MFLOPs) is again four orders of magnitude smaller than the computational work of a single client per communication round ($\approx 7,200,000$ MFLOPs). This server-side step adds no meaningful latency to the overall federated learning process, which is dominated by client-side training and communication.

### A.17 CONVERGENCE ANALYSIS

To analyze the convergence of FedDOR, we first introduce the necessary notations and assumptions. Unless stated otherwise, we denote the client-side loss function as $\mathcal{L}_k(\mathcal{D}_k, \omega_k, \mathcal{P})$, abbreviated as $\mathcal{L}_k$, where $\omega_k = (\phi_k, \theta_k)$ represents the parameters of the feature extractor and classifier of client $k$, and $\mathcal{P}$ denotes the global prototypes.

**Assumption 5 (L-Smoothness).**

For any client $k$, the local loss function $L_k(w_k, P)$ is $L_1$-smooth with respect to its local model parameters $w_k$. That is, for a constant $L_1 > 0$, the following holds:

$$\|\nabla L_k(w_a, P) - \nabla L_k(w_b, P)\| \leq L_1 \|w_a - w_b\|$$

**Assumption 6 (Unbiased Gradient and Bounded Variance).**

The stochastic gradient $g_k(w_k, P)$ computed by client $k$ is an unbiased estimator of the true gradient $\nabla L_k(w_k, P)$, and its variance is bounded by a constant $\sigma^2 \geq 0$:

$$\mathbb{E}[g_k(w_k, P)] = \nabla L_k(w_k, P) \quad \text{and} \quad \mathbb{E}[\|g_k(w_k, P) - \nabla L_k(w_k, P)\|^2] \leq \sigma^2$$

**Assumption 7 (Bounded Server-Side Gradient).**

The expected squared norm of the stochastic gradient for the server-side prototype generator $G$ is bounded by a constant $G^2 > 0$:

$$\mathbb{E}[\|\nabla L_{\text{server}}(\Omega_g)\|^2] \leq G^2$$

**Assumption 8 (Smoothness of Prototype Generator).**

The server-side prototype generator $G(E; \Omega_g)$ is $L_g$-Lipschitz continuous with respect to its parameters $\Omega_g$. That is, for a constant $L_g > 0$:

$$\|G(E; \Omega_a) - G(E; \Omega_b)\| \leq L_g \|\Omega_a - \Omega_b\|$$

**Lemma 4 (Progress in Local Client Training).**

Under Assumptions 5 and 6, after one round of local training consisting of $E$ epochs with a learning rate $\eta < 1/L_1$, the expected local loss function for any client $k$ satisfies:

$$\mathbb{E}[L_k(w_k^{t+1}, P^t)] \leq \mathbb{E}[L_k(w_k^t, P^t)] - C_1 \eta \sum_{e=0}^{E-1} \mathbb{E}[\|\nabla L_k(w_k^{t;e})\|^2] + C_2 \eta^2$$

where $C_1$ and $C_2$ are positive constants. This lemma shows that the local training procedure effectively reduces the client's loss in proportion to the magnitude of its gradients.

**Lemma 5 (Bounded Perturbation from Server Update).**

Under Assumptions 7 and 8, the update of the global prototype from $P^t$ to $P^{t+1}$ introduces a bounded change in the clients' objective functions:

$$\mathbb{E}\left[|L_k(w, P^{t+1}) - L_k(w, P^t)|\right] \leq \delta$$

where $\delta$ is a small positive constant. This lemma is crucial as it ensures that the learning target for the clients remains stable across communication rounds.

**Theorem 2 (Global Convergence of FedDOR).**

Under Assumptions 5-8, by choosing a sufficiently small learning rate $\eta$, the FedDOR algorithm converges to a stationary point. Specifically, the average of the expected squared gradient norms is bounded as follows:

$$\frac{1}{T}\sum_{t=0}^{T-1}\mathbb{E}\left[\|\nabla F(w^t)\|^2\right] \leq \frac{F(w^0) - F^*}{\alpha T} + \beta$$

where $F(w^t)$ is the global loss at round $t$, and $\alpha, \beta$ are positive constants.

The proof follows by applying a telescoping sum to the single-round progress, which is derived from Lemma 4 and Lemma 5. As $T \to \infty$, the first term on the right-hand side vanishes, proving that the algorithm converges to a neighborhood of a stationary point at a rate of $O(1/T)$.

