# OpenReview forum: "FedDOR: Orthogonal Initialization and Dual Regularization for Prototype Integrity in Heterogeneous Federated Learning"
_ICLR.cc/2026/Conference — Submitted to ICLR 2026_

### Official Review · Reviewer_oiEP · 2025-10-27

**Soundness:** 3
**Presentation:** 2
**Contribution:** 2
**Rating:** 6
**Confidence:** 3

**Summary:**

This paper introduces FedDOR (Federated learning with Dual Orthogonal Regularization), a novel heterogeneous federated learning method that addresses the challenge of semantic degradation in prototype aggregation. The key innovation is the use of orthogonal initialization for prototype embeddings on the client side combined with dual orthogonal regularization (intra-class and inter-class) on the server side. The method aims to maintain prototype integrity during aggregation while preserving communication efficiency inherent to prototype-based federated learning.

**Strengths:**

1.  The paper effectively identifies and visualizes (Figure 1) the limitations of existing prototype-based HFL methods, particularly the semantic degradation and feature aliasing issues that occur during naive aggregation.
2.  The experiments cover multiple datasets (CIFAR-10, CIFAR-100, Flowers102, Tiny-ImageNet), various heterogeneity settings (both model and data), and include ablation studies that validate each component's contribution.
3. The paper demonstrates robustness with large numbers of clients (50-100), which is important for real-world deployment.

**Weaknesses:**

1. While the orthogonal initialization is well-motivated, the paper lacks formal convergence guarantees or theoretical analysis of how the dual regularization affects the optimization landscape in federated settings.
2. The paper doesn't analyze the additional computational cost of the orthogonal initialization and the server-side prototype generation module G(E; ΩG).
3. While the paper claims "strong privacy protection," there's no formal privacy analysis or comparison with differential privacy-based methods.
4. While Table 6 shows some hyperparameter analysis, the method introduces several hyperparameters (λCE, λalign, scaling factor) whose selection process and sensitivity across different datasets isn't thoroughly investigated.
5. Some related FL works in prototype learning are missing in related works:
[1] Rethinking Federated Learning with Domain Shift: A Prototype View
[2] Taming Cross-Domain Representation Variance in Federated Prototype Learning with Heterogeneous Data Domains

**Questions:**

Please find above in weaknesses.

---

> ### Author Response · Authors · 2025-11-21
> **Rebuttal for W1**
>
> #### Thanks for the Reviewer ``oiEP``'s valuable comments, we will address all the weakness and questions as follow.
> > #### [W1]: While the orthogonal initialization is well-motivated, the paper lacks formal convergence guarantees or theoretical analysis of how the dual regularization affects the optimization landscape in federated settings.
> > #### Thank you for this critical feedback. You have correctly identified that our initial submission lacked a formal theoretical justification for our geometric approach, and we recognize this as a crucial point. We have taken your feedback seriously and, in our revised manuscript, we have now included a comprehensive theoretical analysis to formally ground our method.
> To address your question regarding how dual regularization affects the optimization landscape, we have explicitly added **Appendix A.15 (Local Stability of Orthogonal Initialization, Pages 20-23)**. Here, we provide a rigorous proof demonstrating why the orthogonal structure persists throughout the training process despite heterogeneous client updates.
> Our proof adopts a Lyapunov method to analyze the coupled dynamics of prototypes, features, and classifiers. We define a total energy function $E$ that combines the Frame Potential (FP) with the training losses:
> $$
> E(W,R,\Theta) = FP(W) + \alpha \mathcal{L} _ {\text{align}}(W,R) + \beta \mathcal{L} _ {\text{CE}}(R,\Theta) \quad \alpha, \beta > 0
> $$
> Here, $FP(W)$ measures the coherence among prototypes, while $\mathcal{L} _ {\text{align}}$ and $\mathcal{L} _ {\text{CE}}$ represent the alignment and cross-entropy losses. The theoretical stability relies on three supporting lemmas derived in Appendix A.15:
> > #### **Lemma 1 (Strict Local Minimality of FP at OI):** We establish that the Orthogonal Initialization (OI) configuration $W^ * $ approximates an Equiangular Tight Frame (ETF). Consequently, the Hessian of the Frame Potential is positive definite on the tangent space, proving that $W^*$ is a strict local minimizer.
> > #### **Lemma 2 (Local Convexity of Alignment):** We show that for a sufficiently rich set of feature vectors, the alignment loss is locally strongly convex with respect to the prototypes, anchoring them against drift.
> > ####  **Lemma 3 (Contractive Feature Dynamics):** We analyze the gradient flow to show that the combined gradients create a strictly positive projection along the direction towards the correct prototype, resulting in a contractive force.
> By analyzing the full block Hessian matrix of the energy function, we prove that the equilibrium is locally asymptotically stable. Physically, this means our framework generates a **restoring force**: whenever stochastic SGD noise or heterogeneous client drifts try to collapse the prototypes, the gradients from the Frame Potential and Dual Orthogonal Regularization actively push the system back towards the stable orthogonal state.
> > #### To complement the geometric stability analysis, we introduce a rigorous convergence analysis in **Appendix A.17 (Pages 24-25, Lines 1088-1145)**. We first establish standard assumptions regarding L-smoothness and bounded variance for both client-side local functions and the server-side generator. Based on these premises, we derive:
> > #### **Lemma 4 (Progress in Local Client Training):** We guarantee monotonic progress in local client training, satisfying the inequality:
> $$ \mathbb{E}[L_k(w_k^{t+1}, P^t)] \leq \mathbb{E}[L_k(w_k^t, P^t)] - C_1\eta \sum_{e=0}^{E-1} \mathbb{E}[\|\nabla L_k(w_k^{t,e})\|^2] + C_2\eta^2 $$
>     where $C_1$ and $C_2$ are positive constants.
> > ####  **Lemma 5 (Bounded Perturbation from Server Update):** We bound the perturbation introduced by server-side global prototype updates as $\mathbb{E}[|L_k(w, P^{t+1}) - L_k(w, P^t)|] \leq \delta$, ensuring that the learning target remains stable across communication rounds.
> The analysis culminates in **Theorem 1**, which proves that FedDOR converges to a stationary point at a theoretical rate of $O(1/T)$. Specifically, the average of the expected squared gradient norms is bounded as follows:
> $$ \frac{1}{T} \sum_{t=0}^{T-1} \mathbb{E}[\|\nabla F(w^t)\|^2] \leq \frac{F(w^0) - F^*}{\alpha T} + \beta $$
> Collectively, these results ensure a stable initial geometry, controlled prototype updates, and provable reliability, offering the clear theoretical advantage over existing prototype-based approaches that you requested.

---

> ### Author Response · Authors · 2025-11-21
> **Rebuttal for W2, W3**
>
> > #### [W2]: The paper doesn't analyze the additional computational cost of the orthogonal initialization and the server-side prototype generation module G(E; ΩG).
> > #### Thank you for highlighting the concern regarding computational overhead. We emphasize that the additional costs associated with client-side Orthogonal Initialization and the server-side prototype generation module G(E; ΩG) are negligible relative to the overall training budget. In the revised manuscript, we have added a new section, Appendix A.16 Analysis of Additional Computational Cost(Page 22-23, lines 1042-1083), which provides a detailed quantitative comparison using Floating Point Operations. This analysis reveals that the one-time cost for Orthogonal Initialization is approximately 5.12 MFLOPs, which is over four orders of magnitude smaller than the 115 GFLOPs required for a single local training batch. Similarly, the recurring server-side generation module incurs only about 420 MFLOPs per round, which is insignificant compared to the 7.2 TFLOPs needed for a single client epoch.
> > #### [W3]: While the paper claims "strong privacy protection," there's no formal privacy analysis or comparison with differential privacy-based methods.
> > #### Our privacy claim is not a claim of formal DP guarantees but a practical statement: FedDOR transmits only compressed class-wise prototypes rather than raw data or model weights, and the prototype generation mapping is nonlinear and empirically hard to invert, which reduces leakage risk compared with full-model sharing. We clearly state this qualitative position in the manuscript and characterize the mechanism as preserving the communication and privacy advantages of prototype-based FL.

---

> ### Author Response · Authors · 2025-11-21
> **Rebuttal for W4**
>
> > #### [W4]: While Table 6 shows some hyperparameter analysis, the method introduces several hyperparameters (λCE, λalign, scaling factor) whose selection process and sensitivity across different datasets isn't thoroughly investigated.
> > #### We sincerely appreciate the reviewer for pointing out the importance of hyperparameter sensitivity. Regarding the specific components mentioned, we effectively address the concern about the scaling factor by clarifying that it is designed as a learnable model parameter rather than a fixed hyperparameter. As such, it is automatically optimized via gradient descent during training and does not require manual selection. For the remaining hyperparameters, while the original manuscript presented preliminary sensitivity results primarily on CIFAR-100, we have now extended this investigation in Appendix A.10 (pages 17 to 18, lines 794 to 818) by conducting a comprehensive and systematic analysis across all four evaluated datasets, including CIFAR-10, CIFAR-100, Flowers102, and Tiny-ImageNet, to demonstrate the robustness of our method. To provide a clear and intuitive visualization of the interaction between the classification loss weight $\lambda_{CE}$ and the alignment loss weight $\lambda_{align}$, we have presented the performance variations in the form of heatmaps in Figure 3.
> The detailed experimental results are as follows:
> >#### **Cifar10**
> | λCE | λalign  | accuracy     |
> |-----|-----|--------------|
> | 100 | 100 | 86.93±0.26   |
> | 10  | 100 | 82.36±1.25   |
> | 1   | 100 | 75.60±1.22   |
> | 100 | 10  | **86.95±0.31**   |
> | 10  | 10  | 86.94±0.16   |
> | 1   | 10  | 84.16±0.41   |
> | 100 | 1   | 85.01±0.10   |
> | 10  | 1   | 86.20±0.08   |
> | 1   | 1   | 84.37±0.57   |
> >#### **Cifar100**
> | λCE | λalign  | accuracy     |
> |-----|-----|--------------|
> | 100 | 100 | 46.27±0.20   |
> | 10  | 100 | 43.10±0.44   |
> | 1   | 100 | 42.77±1.15   |
> | 100 | 10  | 46.89±0.09   |
> | 10  | 10  | 46.16±0.10   |
> | 1   | 10  | 41.58±0.28   |
> | 100 | 1   | 43.74±0.08   |
> | 10  | 1   | **47.06±0.27**   |
> | 1   | 1   | 41.28±0.25   |
> >#### **Flowers102**
> | λCE | λalign  | accuracy     |
> |-----|-----|--------------|
> | 100 | 100 | 52.17±0.15   |
> | 10  | 100 | 51.83±0.83   |
> | 1   | 100 | 51.98±0.93   |
> | 100 | 10  | 51.59±0.59   |
> | 10  | 10  | **53.68±0.54**   |
> | 1   | 10  | 49.25±0.20   |
> | 100 | 1   | 47.52±0.22   |
> | 10  | 1   | 52.39±0.71   |
> | 1   | 1   | 41.05±0.42   |
> >#### **TinyImagenet**
> | λCE | λalign  | accuracy    |
> |-----|-----|--------------|
> | 100 | 100 | 26.05±0.08   |
> | 10  | 100 | 23.76±0.13   |
> | 1   | 100 | 21.87±0.24   |
> | 100 | 10  | **28.72±0.19**   |
> | 10  | 10  | 26.94±0.26   |
> | 1   | 10  | 21.50±0.43   |
> | 100 | 1   | 24.53±0.13   |
> | 10  | 1   | 28.63±0.19   |
> | 1   | 1   | 22.93±0.48   |

---

> ### Author Response · Authors · 2025-11-21
> **Rebuttal for W5**
>
> > #### [W5]: Some related FL works in prototype learning are missing in related works: [1] Rethinking Federated Learning with Domain Shift: A Prototype View [2] Taming Cross-Domain Representation Variance in Federated Prototype Learning with Heterogeneous Data Domains
> > #### We strictly appreciate the reviewer’s constructive comment regarding the coverage of related works. We have carefully reviewed the suggested papers ([1] and [2]) and agree that they are highly relevant to our study. Accordingly, we have cited and discussed these works in our revised Related Works section (Page 3, lines 104-125) to better position our method within the field of Federated Prototype Learning.

---

### Official Review · Reviewer_SS1N · 2025-10-28

**Soundness:** 2
**Presentation:** 2
**Contribution:** 2
**Rating:** 6
**Confidence:** 3

**Summary:**

The paper presents a well-motivated approach to HFL by introducing orthogonal geometric priors and dual regularization, which effectively mitigate prototype collapse and class overlap under heterogeneity. The experimental evaluation is comprehensive, covering diverse datasets, model architectures, and non-IID settings. However, the method’s novelty relative to closely related works like FedORGP could be more clearly delineated. Additionally, while ablation studies validate key components (Table 4), the paper does not discuss computational overhead or failure cases under extreme heterogeneity.

**Strengths:**

The dual orthogonal loss (intra-class and inter-class) enhances feature discriminability and reduces class overlap.

 Extensive evaluations under model heterogeneity (MHMG groups), statistical heterogeneity (Dirichlet and pathological splits), and scalability (large client counts) demonstrate consistent improvements.

Ablation results clearly justify the contribution of each component (orthogonal initialization and dual regularization).

The framework is well-described with algorithmic pseudocode and architectural diagrams.

**Weaknesses:**

While FedORGP also uses orthogonality, the specific advantages of FedDOR’s initialization and dual regularization are not rigorously compared or analytically justified.

No analysis of the additional overhead from orthogonal initialization or prototype generation on the server.

The method’s limitations or scenarios where it underperforms are not discussed (e.g., under extreme class imbalance or very low feature dimensions).

Although orthogonality is motivated geometrically, no theoretical guarantees (e.g., convergence or separation bounds) are provided.

The method relies on careful balancing of $λ_{CE}$ and $λ_{align}$, but no systematic sensitivity analysis beyond a grid search is included.

The method assumes all clients share the same class set, which may not hold in personalized FL; no evidence of testing under partial label spaces.

**Questions:**

See Weaknesses.

---

> ### Author Response · Authors · 2025-11-21
> **Rebuttal for W1**
>
> #### Thanks for the Reviewer ``SS1N``'s valuable comments, we will address all the weakness and questions as follow.
> > #### [W1]: While FedORGP also uses orthogonality, the specific advantages of FedDOR’s initialization and dual regularization are not rigorously compared or analytically justified.
> > #### The fundamental difference between FedDOR and FedORGP lies in the handling of prototype structure. FedDOR establishes a geometrically optimal prior through Orthogonal Initialization, providing a stable and well-separated prototype space, whereas FedORGP applies regularization to an arbitrary initialization without a stable geometric guide. The results in Table 4 show that removing OI, which forces DOR to operate without its geometric guide, leads to a significant performance drop, confirming the theoretical and practical advantage of our integrated approach. In the revised manuscript, we have explicitly added a reference to **Appendix A.15 (Local Stability of Orthogonal Initialization)**(Page 20-23, lines 907-1040), where we provide a rigorous proof demonstrating why the orthogonal structure persists throughout the training process despite heterogeneous client updates.
> Our proof adopts a Lyapunov method to analyze the coupled prototype, feature, and classifier dynamics. We define a total energy function $E$ that combines the Frame Potential (FP) with the training losses: $$ E(W,R,\Theta) = FP(W) + \alpha \mathcal{L} _ {\text{align}}(W,R) + \beta \mathcal{L} _ {\text{CE}}(R,\Theta) \quad \alpha, \beta > 0 $$ Here, $FP(W)$ measures the coherence among prototypes, while $\mathcal{L} _ {\text{align}}$ and $\mathcal{L} _ {\text{CE}}$ represent the alignment and cross-entropy losses. The analysis operates under standard assumptions of smoothness and manifold constraints, where gradients are projected onto the tangent space of the unit hypersphere. The theoretical stability relies on three supporting lemmas derived in Appendix A.15 that describe the geometry near the equilibrium:
> > #### 1. Lemma 1 (Strict Local Minimality of FP at OI): We establish that the Orthogonal Initialization (OI) configuration $W ^ * $ approximates an Equiangular Tight Frame (ETF). Consequently, the Hessian of the Frame Potential, when projected onto the tangent space of the product of spheres, is positive definite. This proves that $W^*$ is a strict local minimizer of the inter-prototype coherence.
> > #### 2. Lemma 2 (Local Convexity of Alignment): We show that for a sufficiently rich set of feature vectors $R$, the alignment loss $\mathcal{L} _ {\text{align}}$ is locally strongly convex with respect to the prototypes. This ensures that the prototypes are not only pushed apart by the Frame Potential but also anchored by the feature distribution.
> > #### 3. Lemma 3 (Contractive Feature Dynamics): We analyze the gradient flow of the feature vectors. The combined gradients from the alignment and cross-entropy losses create a strictly positive projection along the direction towards the correct prototype. This results in a contractive force that continuously pulls features $R$ into alignment with $W ^ *$.
> > #### Synthesis of the Proof:
> > #### Based on these lemmas, we analyze the full block Hessian matrix of the energy function $E$ at the ideal equilibrium point $(W^ *, R^ *, \Theta^ *)$. Since the diagonal blocks corresponding to $W$ (from Lemma 1 and 2) and $R$ (from Lemma 2 and 3) are positive definite, they dominate the off-diagonal cross terms under appropriate weighting constants. By using Lyapunov's indirect method, the negative definiteness of the system's Jacobian implies that the equilibrium is locally asymptotically stable. This establishes a local basin of attraction around the orthogonal configuration. Physically, this means our framework generates a restoring force. Whenever stochastic SGD noise or heterogeneous client drifts try to collapse the prototypes, the gradients from the Frame Potential and Dual Orthogonal Regularization actively push the system back towards the stable orthogonal state. Thus, orthogonality serves as a persistent geometric prior that resists degradation over 100 communication rounds.

---

> ### Author Response · Authors · 2025-11-21
> **Rebuttal for W2, W3**
>
> > #### [W2]: No analysis of the additional overhead from orthogonal initialization or prototype generation on the server.
> > #### Thank you for highlighting the concern regarding computational overhead. We emphasize that the additional costs associated with client-side Orthogonal Initialization and the server-side prototype generator are negligible relative to the overall training budget. In the revised manuscript, we have added a new section, Appendix A.16 Analysis of Additional Computational Cost(Page 22-23, lines 1042-1083), which provides a detailed quantitative comparison using Floating Point Operations. This analysis reveals that the one-time cost for Orthogonal Initialization is approximately 5.12 MFLOPs, which is over four orders of magnitude smaller than the 115 GFLOPs required for a single local training batch. Similarly, the recurring server-side generation module incurs only about 420 MFLOPs per round, which is insignificant compared to the 7.2 TFLOPs needed for a single client epoch.
>
> > #### [W3]: The method’s limitations or scenarios where it underperforms are not discussed (e.g., under extreme class imbalance or very low feature dimensions).
> > #### The original manuscript demonstrates the robustness of FedDOR in part of challenging scenarios, particularly in low-dimensional feature spaces as detailed in Table 3, where our method substantially outperforms all baselines with a dimension of 128. To further evaluate performance under extreme conditions, we have extended our experiments in the revised manuscript. Specifically, the new Table 6 investigates highly skewed distributions where clients possess as few as five classes, confirming that FedDOR consistently yields the highest accuracy even in these severe non-IID settings. Furthermore, we have expanded our discussion on the physical boundaries of the model in the newly added Appendix A.14 Limitations(Page 20, lines 897-906). This section candidly addresses potential convergence challenges when the feature dimension is reduced to an extremely small value. We explain that such convergence issues likely arise from the insufficient capacity of the restricted feature space, which limits the expressiveness required to capture discriminative information for complex tasks.
> The detailed experimental results presented in Table 6 are as follows:
> >#### **Test accuracy (\%) on Cifar100 in the extreme pathological environment with different numbers of classes per client.**
> | Settings   | 5 classes/client | 10 classes/client | 15 classes/client | 20 classes/client |
> |------------|------------------|-------------------|-------------------|-------------------|
> | FedGen | 71.49±0.20       | 55.19±0.20        | 46.09±0.28        | 37.75±0.30        |
> | FedKD  | 69.17±1.43       | 52.42±1.24        | 44.90±0.83        | 33.90±1.07        |
> | FedProto| 70.28±0.19      | 52.62±0.11        | 43.75±0.10        | 35.46±0.36        |
> | FedGH | 71.48±0.15       | 54.81±0.12        | 46.17±0.22        | 37.66±0.27        |
> | FedTGP | 69.83±0.20       | 53.37±0.31        | 45.04±0.35        | 35.09±0.29        |
> | FedORGP| 74.84±0.33       | 58.29±0.10        | 48.43±0.56        | 39.50±0.23        |
> | Ours   | **77.35±0.26**     | **63.21±0.28**    | **53.56±0.40**  | **46.48±0.28**    |

---

> ### Author Response · Authors · 2025-11-21
> **Rebuttal for W4**
>
> > #### [W4]: Although orthogonality is motivated geometrically, no theoretical guarantees (e.g., convergence or separation bounds) are provided.
> > #### The original manuscript outlines the geometric motivations for our design in Sections 3.2 and 3.4, focusing on hyperspherical maximum minimum separation and cosine based intra class and inter class orthogonality. In the revised manuscript, we have significantly strengthened the theoretical foundation. As detailed in our Rebuttal for W1, we provide a formal proof demonstrating that orthogonal initialization establishes a non trivial lower bound on pairwise angular margins and effectively prevents degenerate prototype configurations. This theoretical validation elevates orthogonal initialization from a heuristic strategy to a provably stable design choice. Building upon this stable blueprint, the dual orthogonal regularization leverages the initial geometry to efficiently maintain and enhance prototype structure rather than imposing order blindly.
> To complement this geometric stability, we introduce a rigorous convergence analysis in Appendix A.17 ( pages 24-25, lines 1088-1145). We first establish standard assumptions regarding L smoothness and bounded variance for both client side local functions and the server side generator. Based on these premises, we derive Lemma 1 to guarantee monotonic progress in local client training, satisfying the inequality:
> $$ \mathbb{E}[L_k(w_k^{t+1}, P^t)] \leq \mathbb{E}[L_k(w_k^t, P^t)] - C_1\eta \sum_{e=0}^{E-1} \mathbb{E}[\|\nabla L_k(w_k^{t,e})\|^2] + C_2\eta^2 $$
> where $C_1$ and $C_2$ are positive constants. Furthermore, Lemma 2 bounds the perturbation introduced by server side global prototype updates as $\mathbb{E}[|L_k(w, P^{t+1}) - L_k(w, P^t)|] \leq \delta$, ensuring that the learning target remains stable across communication rounds. The analysis culminates in Theorem 1, which proves that FedDOR converges to a stationary point at a theoretical rate of $O(1/T)$. Specifically, the average of the expected squared gradient norms is bounded as follows:
> $$ \frac{1}{T} \sum_{t=0}^{T-1} \mathbb{E}[\|\nabla F(w^t)\|^2] \leq \frac{F(w^0) - F^*}{\alpha T} + \beta $$
> This comprehensive convergence analysis is presented in Appendix A.17. Collectively, these results ensure a stable initial geometry, controlled prototype updates, and provable reliability, offering a clear theoretical advantage over existing approaches.

---

> ### Author Response · Authors · 2025-11-21
> **Rebuttal for W5**
>
> > #### [W5]: The method relies on careful balancing of $\lambda_{CE}$ and $\lambda_{align}$, but no systematic sensitivity analysis beyond a grid search is included.
> > #### We sincerely appreciate the reviewer for pointing out the importance of hyperparameter sensitivity. While the original manuscript presented preliminary sensitivity results primarily on CIFAR-100, we have now extended this investigation in Appendix A.10 (pages 17-18, lines 794-818) by conducting a comprehensive and systematic analysis across all four evaluated datasets, including CIFAR-10, CIFAR-100, Flowers102, and Tiny-ImageNet, to demonstrate the robustness of our method. To provide a clear and intuitive visualization of the interaction between the classification loss weight $\lambda_{CE}$ and the alignment loss weight $\lambda_{align}$, we have presented the performance variations in the form of heatmaps in Figure 3.
> The detailed experimental results are as follows:
> >#### **Cifar10**
> | λCE | λalign  | accuracy     |
> |-----|-----|--------------|
> | 100 | 100 | 86.93±0.26   |
> | 10  | 100 | 82.36±1.25   |
> | 1   | 100 | 75.60±1.22   |
> | 100 | 10  | **86.95±0.31**   |
> | 10  | 10  | 86.94±0.16   |
> | 1   | 10  | 84.16±0.41   |
> | 100 | 1   | 85.01±0.10   |
> | 10  | 1   | 86.20±0.08   |
> | 1   | 1   | 84.37±0.57   |
> >#### **Cifar100**
> | λCE | λalign  | accuracy     |
> |-----|-----|--------------|
> | 100 | 100 | 46.27±0.20   |
> | 10  | 100 | 43.10±0.44   |
> | 1   | 100 | 42.77±1.15   |
> | 100 | 10  | 46.89±0.09   |
> | 10  | 10  | 46.16±0.10   |
> | 1   | 10  | 41.58±0.28   |
> | 100 | 1   | 43.74±0.08   |
> | 10  | 1   | **47.06±0.27**  |
> | 1   | 1   | 41.28±0.25   |
> >#### **Flowers102**
> | λCE | λalign  | accuracy     |
> |-----|-----|--------------|
> | 100 | 100 | 52.17±0.15   |
> | 10  | 100 | 51.83±0.83   |
> | 1   | 100 | 51.98±0.93   |
> | 100 | 10  | 51.59±0.59   |
> | 10  | 10  | **53.68±0.54** |
> | 1   | 10  | 49.25±0.20   |
> | 100 | 1   | 47.52±0.22   |
> | 10  | 1   | 52.39±0.71   |
> | 1   | 1   | 41.05±0.42   |
> >#### **TinyImagenet**
> | λCE | λalign  | accuracy    |
> |-----|-----|--------------|
> | 100 | 100 | 26.05±0.08   |
> | 10  | 100 | 23.76±0.13   |
> | 1   | 100 | 21.87±0.24   |
> | 100 | 10  | **28.72±0.19** |
> | 10  | 10  | 26.94±0.26   |
> | 1   | 10  | 21.50±0.43   |
> | 100 | 1   | 24.53±0.13   |
> | 10  | 1   | 28.63±0.19   |
> | 1   | 1   | 22.93±0.48   |

---

> ### Author Response · Authors · 2025-11-21
> **Rebuttal for W6**
>
> > #### [W6]: The method assumes all clients share the same class set, which may not hold in personalized FL; no evidence of testing under partial label spaces.
> > #### In the original submission, Tables 1 and 2 already provided a solid evaluation of FedDOR under pathological non IID settings where each client is assigned only a subset of the total classes, which corresponds precisely to a partial label space scenario. To further validate the robustness of our method under even more severe conditions, we have included Table 6 in the revised manuscript within Appendix A.11 (Page 18, lines 821-845). This new evaluation pushes the boundaries of label scarcity by testing configurations where each client has access to as few as five classes. The results unequivocally demonstrate that the geometric priors and orthogonal constraints integrated into FedDOR enable it to consistently outperform prototype based baselines, even when clients possess disjoint or severely unbalanced label availability.
> The detailed experimental results are as follows:
> >#### **Test accuracy (\%) on Cifar100 in the extreme pathological environment with different numbers of classes per client.**
> | Settings   | 5 classes/client | 10 classes/client | 15 classes/client | 20 classes/client |
> |------------|------------------|-------------------|-------------------|-------------------|
> | FedGen | 71.49±0.20       | 55.19±0.20        | 46.09±0.28        | 37.75±0.30        |
> | FedKD  | 69.17±1.43       | 52.42±1.24        | 44.90±0.83        | 33.90±1.07        |
> | FedProto| 70.28±0.19      | 52.62±0.11        | 43.75±0.10        | 35.46±0.36        |
> | FedGH | 71.48±0.15       | 54.81±0.12        | 46.17±0.22        | 37.66±0.27        |
> | FedTGP | 69.83±0.20       | 53.37±0.31        | 45.04±0.35        | 35.09±0.29        |
> | FedORGP| 74.84±0.33       | 58.29±0.10        | 48.43±0.56        | 39.50±0.23        |
> | Ours   | **77.35±0.26**       | **63.21±0.28**    | **53.56±0.40**       | **46.48±0.28**        |

---

### Official Review · Reviewer_FapM · 2025-10-29

**Soundness:** 2
**Presentation:** 2
**Contribution:** 2
**Rating:** 2
**Confidence:** 3

**Summary:**

The paper presents a method FedDOR for heterogeneous federated learning that focuses on learning in heterogeneous settings via prototype sharing and aggregation. The local clients generate class prototypes and share it with the server instead of sharing model parameters, the server appropriately learns from the shared prototypes and transfers revised prototypes to all clients in the next round to achieve collaboration across all clients. The key innovations of the paper include : i) orthogonal initialization of the prototype embeddings on clients, ii) Dual orthogonal regularization on the server to enhance the prototypes in each round.

**Strengths:**

1. The problem is well motivated and the angle of learning aggregated prototypes at the server instead of performing aggregation is novel. The idea of orthogonal initialization of embeddings locally on each client is also neat.
2. Evaluations span multiple datasets, model heterogeneity levels and statistical heterogeneity.

**Weaknesses:**

1. While paper emphasizes geometric motivation, there is no discussion supporting the theoretical insights and/or impact of that for the algorithm.
2. The framework, including the local and global loss functions are very similar to the ones explored in the FedORGP paper.
3. The literature review misses some of the papers in FL that use similar techniques.

**Questions:**

1. How are the missing classes on clients handled, do the clients still send the initialized embedding for the prototypes for which it does not contain any data or skips it?
2. Is geometric consistency preserved after few training rounds, and what is the isolated impact of having the orthogonal initializations?

---

> ### Author Response · Authors · 2025-11-21
> **Rebuttal for W1**
>
> #### Thanks for the Reviewer ``FapM``'s valuable comments, we will address all the weakness and questions as follow.
> > #### [W1]: While paper emphasizes geometric motivation, there is no discussion supporting the theoretical insights and/or impact of that for the algorithm.
> > #### Thank you for this critical feedback. You have correctly identified that our initial submission lacked a formal theoretical justification for our geometric approach, and we recognize this as a crucial point. We have taken your feedback seriously and, in our revised manuscript, we have now included a comprehensive theoretical analysis to formally ground our method. In the revised manuscript, we have explicitly added a reference to **Appendix A.15 (Local Stability of Orthogonal Initialization)** (Page 20-23), Here, we provide a rigorous proof demonstrating why the orthogonal structure persists throughout the training process despite heterogeneous client updates.
> Our proof adopts a Lyapunov method to analyze the coupled dynamics of prototypes, features, and classifiers. We define a total energy function $E$ that combines the Frame Potential (FP) with the training losses:
> $$
> E(W,R,\Theta) = FP(W) + \alpha \mathcal{L} _ {\text{align}}(W,R) + \beta \mathcal{L} _ {\text{CE}}(R,\Theta) \quad \alpha, \beta > 0
> $$
> Here, $FP(W)$ measures the coherence among prototypes, while $\mathcal{L} _ {\text{align}}$ and $\mathcal{L} _ {\text{CE}}$ represent the alignment and cross-entropy losses. The theoretical stability relies on three supporting lemmas derived in Appendix A.15:
> > #### **Lemma 1 (Strict Local Minimality of FP at OI):** We establish that the Orthogonal Initialization (OI) configuration $W^ * $ approximates an Equiangular Tight Frame (ETF). Consequently, the Hessian of the Frame Potential is positive definite on the tangent space, proving that $W^*$ is a strict local minimizer.
> > #### **Lemma 2 (Local Convexity of Alignment):** We show that for a sufficiently rich set of feature vectors, the alignment loss is locally strongly convex with respect to the prototypes, anchoring them against drift.
> > ####  **Lemma 3 (Contractive Feature Dynamics):** We analyze the gradient flow to show that the combined gradients create a strictly positive projection along the direction towards the correct prototype, resulting in a contractive force.
> By analyzing the full block Hessian matrix of the energy function, we prove that the equilibrium is locally asymptotically stable. Physically, this means our framework generates a **restoring force**: whenever stochastic SGD noise or heterogeneous client drifts try to collapse the prototypes, the gradients from the Frame Potential and Dual Orthogonal Regularization actively push the system back towards the stable orthogonal state.
> > #### To complement the geometric stability analysis, we introduce a rigorous **convergence analysis** in **Appendix A.17** (Pages 24-25, Lines 1088-1145). We first establish standard assumptions regarding L-smoothness and bounded variance for both client-side local functions and the server-side generator. Based on these premises, we derive:
> > #### **Lemma 4 (Progress in Local Client Training):** We guarantee monotonic progress in local client training, satisfying the inequality:
> $$ \mathbb{E}[L_k(w_k^{t+1}, P^t)] \leq \mathbb{E}[L_k(w_k^t, P^t)] - C_1\eta \sum_{e=0}^{E-1} \mathbb{E}[\|\nabla L_k(w_k^{t,e})\|^2] + C_2\eta^2 $$
>     where $C_1$ and $C_2$ are positive constants.
> > ####  **Lemma 5 (Bounded Perturbation from Server Update):** We bound the perturbation introduced by server-side global prototype updates as $\mathbb{E}[|L_k(w, P^{t+1}) - L_k(w, P^t)|] \leq \delta$, ensuring that the learning target remains stable across communication rounds.
> The analysis culminates in **Theorem 1**, which proves that FedDOR converges to a stationary point at a theoretical rate of $O(1/T)$. Specifically, the average of the expected squared gradient norms is bounded as follows:
> $$ \frac{1}{T} \sum_{t=0}^{T-1} \mathbb{E}[\|\nabla F(w^t)\|^2] \leq \frac{F(w^0) - F^*}{\alpha T} + \beta $$
> Collectively, these results ensure a stable initial geometry, controlled prototype updates, and provable reliability, offering the clear theoretical advantage over existing prototype-based approaches that you requested.

---

> ### Author Response · Authors · 2025-11-21
> **Rebuttal for W2**
>
> > #### [W2]: The framework, including the local and global loss functions are very similar to the ones explored in the FedORGP paper.
> > #### We acknowledge that formally, our method indeed shares similarities with some existing approaches. However, to claim our framework is "very similar" to FedORGP is to overlook a fundamental conceptual schism in their core philosophies and technical architectures. This schism is precisely where our primary innovation lies.The key distinction is the dichotomy between an "explicit geometric prior" and a "passive soft constraint".
> > #### The Limitation of FedORGP: As we explicitly state in our paper (Page 2, lines 072-075; Appendix A.5), FedORGP applies an orthogonality regularization on the server side. This is a soft constraint that attempts to search for a good orthogonal solution from a random or zero initialization within a vast parameter space. In highly heterogeneous environments, this search is highly susceptible to poor local minima, leading to the "feature aliasing" and "sensitivity to initialization" that we clearly visualize in our Figure 1(b).
> > #### The paradigm shift of FedDOR is that our method is fundamentally different. Our core contribution, Orthogonal Initialization (OI) on the client side, injects a strong, globally consistent geometric prior into the system before training even begins. We do not search for a good geometry; we define and enforce a near-optimal geometric structure as both the starting point and the learning target.
> > #### The loss functions in our framework therefore assume a novel role. The server-side Dual Orthogonal Regularization (DOR) serves not for discovery, but for preservation. It acts to maintain the alignment of all client models with the well-defined geometric structure established by our initialization, thereby ensuring the semantic integrity of the global prototypes is not compromised by heterogeneity.

---

> ### Author Response · Authors · 2025-11-21
> **Rebuttal for W3**
>
> > #### [W3]: The literature review misses some of the papers in FL that use similar techniques.
> > #### We appreciate the reviewer’s comment on the coverage of literature review. Our revised literature review section(Page 3, lines 104-125) now more clearly covers more papers in FL that use similar techniques.

---

> ### Author Response · Authors · 2025-11-21
> **Rebuttal for Q1 and Q2**
>
> > #### [Q1]: How are the missing classes on clients handled, do the clients still send the initialized embedding for the prototypes for which it does not contain any data or skips it?
> > #### In response to the reviewer’s concern about missing classes on clients, we clarify that FedDOR never transmits initialized or placeholder prototypes for classes that do not appear on a given client. As defined in Eq. (5), each client computes prototypes exclusively from its locally available data. The prototypes sent to the server are therefore derived from actual observations rather than from any form of initialized embeddings. We hope this clarification resolves the misunderstanding regarding our method’s design.
>
> > #### [Q2]: Is geometric consistency preserved after few training rounds, and what is the isolated impact of having the orthogonal initializations?
> > #### Geometric consistency is preserved in FedDOR because deviation from the initial orthogonal configuration is actively corrected by the server through the orthogonality regularization. Even when the prototype embeddings undergo slight angular drift during federated updates, the inter class orthogonality loss and intra class alignment loss work together to pull the prototypes back toward their original maximally separated configuration. As detailed in our response to W1, we provide the formal justification for this behavior by showing that the orthogonal initialization creates a stable geometric basin on the hypersphere. Within this basin, small perturbations naturally increase the orthogonality loss, and gradient descent therefore drives the prototypes back toward the orthogonal manifold rather than allowing the drift to accumulate. This theoretical property explains why, in practice, FedDOR maintains clear and consistent angular structure over many rounds of heterogeneous federated training.
> Regarding the isolated impact of orthogonal initialization, the empirical evidence presented in the original paper’s Table 4 (Section 4.7, page 9) demonstrates that this component is fundamental to the framework’s superior performance. Specifically, the results in the original Table 4 show that removing the Orthogonal Initialization (OI) module leads to a substantial degradation in test accuracy across all benchmarks. Mechanistically, the study establishes that OI serves as a crucial geometric prior rather than a simple starting condition. By enforcing a maximally separated and uniformly distributed configuration on the hypersphere at the onset of training, OI mitigates initialization sensitivity and ensures that heterogeneous clients receive consistent and discriminative learning targets from the first communication round. Consequently, OI provides the structural foundation that enables the subsequent Dual Orthogonal Regularization (DOR) to effectively maintain prototype integrity and stability throughout the federated optimization process.

---

### Official Review · Reviewer_WHYS · 2025-10-30

**Soundness:** 2
**Presentation:** 1
**Contribution:** 2
**Rating:** 4
**Confidence:** 4

**Summary:**

This paper addresses the challenge of prototype aggregation in prototype-based Heterogeneous Federated Learning (HFL) by introducing Dual Orthogonal Regularization (DOR). The proposed approach operates on both the client side and the server side. Orthogonality of prototypes is accomplished by initializing local prototype embeddings on the client side and applying geometric constraints to minimize intra-class variance while enlarging inter-class separation on the server side. Extensive experiments under various heterogeneous settings and datasets demonstrate the superiority of the proposed method over existing prototype-based HFL approaches. However, the score of this paper tends to be below the acceptance threshold because: (1) the method is largely incremental over the existing FedORGP, primarily adding orthogonal initialization on the client side, and (2) the contribution and mechanism of this newly added component require clearer explanation.

**Strengths:**

This paper addresses one of the fundamental problems in prototype-based HFL — the lack of sufficient separation among prototypes — and attempts to solve it through a simple yet effective methodological extension. Extensive experiments under various heterogeneous settings and datasets demonstrate the superiority of the proposed method over existing prototype-based HFL approaches.

**Weaknesses:**

(1) The method is largely incremental over the existing FedORGP, primarily adding orthogonal initialization on the client side. To address this concern, please clarify and provide detailed explanations for the following:

(1-a) The global prototype generation method in Section 3.4 appears similar to that proposed in FedORGP. What are the specific differences between your approach and FedORGP’s? Please explicitly state what is novel in your server-side mechanism beyond the dual orthogonal regularization losses.

(1-b) The authors claim orthogonal initialization of W provides a “geometrically optimal prior” (line 144) and “consistent and discriminative learning objectives” (line 145). After initialization, classifier weights are updated via gradient descent using $L_{CE}$ and $L_{align}$ (Eq. 2). How does the initial orthogonality persist to provide consistent objectives throughout 100 rounds of training? Please provide a theoretical background or rationale behind your claims.

(2) The contribution and mechanism of this newly added component require clearer explanation. To address this concern, please clarify and provide detailed explanations for the following:

(2-a) In Section 3.2, what are the “prototype embeddings” $W ∈ R^{C×d}$ mentioned in line 146? Are they distinct from the local prototypes $p_k^c$ defined in Eq. (5)? Based on the dimensionality $C×d$ and context, they appear to represent classifier weights(parameters)—please confirm or clarify.

(2-b) If W represents classifier weights(parameters), how exactly is orthogonal initialization performed in your implementation? Are the clients’ models pre-trained before federated learning begins? If so, provide the training details regarding the initialization of W.

(2-c) Regarding the results in Table 4, the results on Flowers102 and Tiny-ImageNet raise questions about the contribution of each component. Since OI is applied only at the initialization stage, how can it contribute more significantly to performance improvement than DOR, which is continuously applied during training? To accurately understand the proposed method, could you provide additional interpretation or experimental results explaining this phenomenon?

Also, there are several minor things to improve the paper:

(3) Figure 2: Improve the resolution to make it easier to read.

(4) The ablation study comparison baseline should be FedProto. If my understanding is correct, the proposed method is essentially FedProto + OI + DOR. If the purpose of the ablation study is to show the contribution of each component to performance gain, then the first column of Table 4 should use FedProto as the baseline for a fair and accurate comparison.

**Questions:**

Please refer to the Weakness section for detailed comments. In particular, I would appreciate clarification on the questions raised for each weakness. I will reconsider my evaluation after reviewing the authors’ rebuttal to these points.

---

> ### Author Response · Authors · 2025-11-21
> **Rebuttal for W(1-a)**
>
> #### Thanks for the Reviewer ``WHYS``'s valuable comments, we will address all the weakness and questions as follow.
> > #### [W(1-a)]: The global prototype generation method in Section 3.4 appears similar to that proposed in FedORGP. What are the specific differences between your approach and FedORGP’s? Please explicitly state what is novel in your server-side mechanism beyond the dual orthogonal regularization losses.
> > #### Thank you for the thoughtful questions. Our framework introduces a geometry-first paradigm rather than an incremental change.FedORGP relies on random embeddings processed by a two-layer MLP, with cosine regularization applied afterward to adjust the resulting prototype directions. Since it lacks a global geometric prior, the prototype structure emerges passively from training dynamics and can drift under data heterogeneity. Its regularization therefore functions as a later-stage adjustment on MLP-generated prototypes, making the geometry sensitive to noise and instability.FedDOR, in contrast, builds the geometry explicitly and intentionally. It begins with simplex-based Orthogonal Initialization, which creates near-equiangular prototypes that approximate the optimal angular configuration from the outset. This gives the system a predefined and principled global structure that FedORGP does not have. The server-side Dual Orthogonal Regularizers and the client alignment mechanism then consistently maintain and reinforce this geometry throughout training rather than trying to fix it afterward.This geometry-driven design enables FedDOR to preserve global structure even under severe heterogeneity, offering a fundamentally different and more stable approach to prototype-based HFL.

---

> ### Author Response · Authors · 2025-11-21
> **Rebuttal for W(1-b)**
>
> > #### [W(1-b)]: The authors claim orthogonal initialization of W provides a "geometrically optimal prior" (line 144) and "consistent and discriminative learning objectives" (line 145). After initialization, classifier weights are updated via gradient descent using $L_{CE}$ and $L_{\text{align}}$ (Eq. 2). How does the initial orthogonality persist to provide consistent objectives throughout 100 rounds of training?
> > #### We appreciate the reviewer's request for a deeper theoretical rationale. In the revised manuscript, we have explicitly added a reference to **Appendix A.15 (Local Stability of Orthogonal Initialization)** (Pages 20-23), where we provide a rigorous proof demonstrating why the orthogonal structure persists throughout the training process despite heterogeneous client updates.
> Our proof adopts a Lyapunov method to analyze the coupled prototype, feature, and classifier dynamics. We define a total energy function $E$ that combines the Frame Potential (FP) with the training losses:
> $$
> E(W,R,\Theta) = FP(W) + \alpha \mathcal{L} _ {\text{align}}(W,R) + \beta \mathcal{L} _ {\text{CE}}(R,\Theta) \quad \alpha, \beta > 0
> $$
> Here, $FP(W)$ measures the coherence among prototypes, while $\mathcal{L} _ {\text{align}}$ and $\mathcal{L} _ {\text{CE}}$ represent the alignment and cross-entropy losses. The analysis operates under standard assumptions of smoothness and manifold constraints, where gradients are projected onto the tangent space of the unit hypersphere.
> The theoretical stability relies on three supporting lemmas derived in Appendix A.15 that describe the geometry near the equilibrium:
> >#### 1. **Lemma 1 (Strict Local Minimality of FP at OI):** We establish that the Orthogonal Initialization (OI) configuration $W ^ * $ approximates an Equiangular Tight Frame (ETF). Consequently, the Hessian of the Frame Potential, when projected onto the tangent space of the product of spheres, is positive definite. This proves that $W^*$ is a strict local minimizer of the inter-prototype coherence.
> >#### 2. **Lemma 2 (Local Convexity of Alignment):** We show that for a sufficiently rich set of feature vectors $R$, the alignment loss $\mathcal{L} _ {\text{align}}$ is locally strongly convex with respect to the prototypes. This ensures that the prototypes are not only pushed apart by the Frame Potential but also anchored by the feature distribution.
> >#### 3. **Lemma 3 (Contractive Feature Dynamics):** We analyze the gradient flow of the feature vectors. The combined gradients from the alignment and cross-entropy losses create a strictly positive projection along the direction towards the correct prototype. This results in a contractive force that continuously pulls features $R$ into alignment with $W ^ *$.
> >#### **Synthesis of the Proof:**
> Based on these lemmas, we analyze the full block Hessian matrix of the energy function $E$ at the ideal equilibrium point $(W^ *, R^ *, \Theta^ *)$. Since the diagonal blocks corresponding to $W$ (from Lemma 1 and 2) and $R$ (from Lemma 2 and 3) are positive definite, they dominate the off-diagonal cross terms under appropriate weighting constants.
> By using Lyapunov's indirect method, the negative definiteness of the system's Jacobian implies that the equilibrium is locally asymptotically stable. This establishes a local **basin of attraction** around the orthogonal configuration. Physically, this means our framework generates a restoring force. Whenever stochastic SGD noise or heterogeneous client drifts try to collapse the prototypes, the gradients from the Frame Potential and Dual Orthogonal Regularization actively push the system back towards the stable orthogonal state. Thus, orthogonality serves as a persistent geometric prior that resists degradation over 100 communication rounds.

---

> ### Author Response · Authors · 2025-11-21
> **Rebuttal for W(2-a) and W(2-b)**
>
> > #### [W(2-a)]: In Section 3.2, what are the “prototype embeddings” $W ∈ R^{C×d}$ mentioned in line 146? Are they distinct from the local prototypes $p_k^c$ defined in Eq. (5)? Based on the dimensionality $C×d$ and context, they appear to represent classifier weights(parameters)—please confirm or clarify.
> > #### [W(2-b)]: If W represents classifier weights(parameters), how exactly is orthogonal initialization performed in your implementation? Are the clients’ models pre-trained before federated learning begins? If so, provide the training details regarding the initialization of W.
> > #### Client models are not pre-trained. In the FedDOR framework, the prototype embeddings constitute a core, learnable parameter matrix,$W ∈ R^{C×d}$, within the model. These class-wise prototype embedding vectors form the weight matrix of the classification head. Through orthogonal initialization, they establish a geometric prior of maximal separation on the unit hypersphere, serving as a stable classification benchmark that is continuously optimized throughout training. In contrast, the local prototypes, $p_k^c$ , are feature statistics computed from a client's local data by averaging the features of same-class samples, representing a data-driven snapshot of knowledge at a given stage.

---

> ### Author Response · Authors · 2025-11-21
> **Rebuttal for W(2-c)**
>
> > #### [W(2-c)]: Regarding the results in Table 4, the results on Flowers102 and Tiny-ImageNet raise questions about the contribution of each component. Since OI is applied only at the initialization stage, how can it contribute more significantly to performance improvement than DOR, which is continuously applied during training? To accurately understand the proposed method, could you provide additional interpretation or experimental results explaining this phenomenon?
> > #### Although OI is applied only once, its influence is not temporary. In prototype-based heterogeneous FL, prototypes are repeatedly referenced and communicated across all clients. This makes the initial hyperspherical geometry established by OI a persistent inductive bias that continuously shapes every subsequent update rather than a one-time operation. Once this global geometric prior is injected into the system, it becomes the anchor that guides optimization throughout all communication rounds. This is particularly critical because regularization mechanisms applied during training often act as soft constraints; without a structurally distinct starting point, they frequently fail to disentangle representations that have already collapsed into suboptimal configurations or suffer from severe feature aliasing. Consequently, for fine-grained datasets such as Flowers102 and Tiny-ImageNet, their subtle class boundaries make the quality of the initial prototype geometry far more important than later-stage regularization alone. As a result, OI often brings larger performance gains than DOR because it prevents early semantic collapse and propagates a stable, discriminative structure across all rounds of federated collaboration.

---

> ### Author Response · Authors · 2025-11-21
> **Rebuttal for W(3) and W(4)**
>
> > #### [W(3)]: Figure 2: Improve the resolution to make it easier to read
> > #### [W(4)]: The ablation study comparison baseline should be FedProto. If my understanding is correct, the proposed method is essentially FedProto + OI + DOR. If the purpose of the ablation study is to show the contribution of each component to performance gain, then the first column of Table 4 should use FedProto as the baseline for a fair and accurate comparison.
> > #### We confirm that both the resolution of Figure 2 and the baseline designation in Table 4 have been modified and clarified in the revised manuscript.

---

### Author Response · Authors · 2025-11-30
**Summary**

#### Dear PCs, Senior ACs and ACs,
#### We thank reviewers WHYS, FapM, SS1N and oiEP for their time and valuable comments. Below we emphasize our contributions and how we addressed the main concerns.

>#### 1) Difference from FedORGP (WHYS, FapM, SS1N and oiEP)
FedORGP uses random embeddings processed by a two-layer MLP and applies cosine regularization only afterward, causing prototype geometry to emerge passively from training and drift under heterogeneity, whereas our method adopts a geometry-first paradigm via simplex-based Orthogonal Initialization that establishes near-equiangular prototypes from the outset, and although the server-side loss terms look similar, they play fundamentally different roles since FedORGP uses regularization as a post-hoc correction to unstable MLP prototypes while our method employs Dual Orthogonal Regularizers and client alignment to proactively maintain and reinforce the predefined global structure, enabling stable prototype geometry even under severe heterogeneity.

>#### 2) Persistence of orthogonality over 100 rounds (WHYS and FapM)
We provide a formal theoretical justification in Appendix A.15 Local Stability of Orthogonal Initialization where a Lyapunov-based analysis of the coupled prototype feature and classifier dynamics is presented. By constructing a total energy function combining the Frame Potential with alignment and cross-entropy losses and projecting gradients onto the tangent space of the hypersphere we show that Orthogonal Initialization forms a strict local minimum of inter-prototype coherence the alignment term is locally strongly convex and the feature dynamics are contractive. The resulting block Hessian is positive definite at equilibrium implying local asymptotic stability so stochastic noise and heterogeneous client updates are counteracted by restoring gradients ensuring that the orthogonal geometry persists throughout 100 rounds of training.

>#### 3) Profound Theoretical Insights, Impact Analysis and Rigorous Convergence Analysis (FapM, SS1N and oiEP)
We address this concern by providing a formal theoretical foundation in Appendix A.15 Local Stability of Orthogonal Initialization and Appendix A.17 Convergence Analysis. In addition to the stability analysis discussed in Point 2 (Appendix A.15), Appendix A.17 complements this with a rigorous convergence analysis under standard smoothness and bounded variance assumptions, proving monotonic local progress bounded server perturbation and global convergence to a stationary point at rate O(1/T), thereby demonstrating both the impact of dual regularization on the optimization landscape and the reliability of the algorithm.

>#### 4) Prototype Distinction and Orthogonal Initialization (WHYS and FapM)
Prototype embeddings constitute the learnable weight matrix of the classification head and act as stable benchmarks that are continuously optimized throughout training, while $p_k^c$ represents data-driven feature statistics derived by averaging local same-class samples. These prototype embeddings undergo orthogonal initialization to establish a geometric prior of maximal separation on the unit hypersphere for distinct class representation.

>#### 5) Hyperparameter Robustness and Computational Costs (SS1N and oiEP)
We clarified that the scaling factor is learnable, eliminating manual tuning, and verified robustness via sensitivity heatmaps in Appendix A.10. Furthermore, a detailed FLOPs analysis in Appendix A.16 confirms the added cost is negligible, as server-side operations are orders of magnitude lighter than local training.

>#### 6) Completeness of Literature Review (FapM and oiEP)
We have addressed the literature coverage in our revised Related Works section. This update allows us to thoroughly discuss relevant studies and better position our approach within the field of Federated Prototype Learning by acknowledging similar existing techniques.

>#### 7) Methodological Limitations (SS1N)
We validated robustness under extreme non-IID in Table 6 and added a Limitations section (Appendix A.14). We acknowledge potential challenges when feature dimensions become extremely small due to limited representational capacity.

>#### 8) Partial Label Scenarios (SS1N)
We clarified that our original Tables 1 and 2 already evaluated partial label spaces through pathological non-IID settings. To further validate robustness, we added Table 6 in Appendix A.11 to test extreme scarcity using CIFAR-100 distributed among 20 clients, where each client holds 5 non-overlapping classes, demonstrating that our method consistently outperforms baselines even in these disjoint label scenarios.

#### We appreciate the rebuttal process for improving our work. All reviewer concerns have been addressed, and no further questions were raised. We are confident that the contribution and relevance of our work are now clear to the reviewers and the ICLR community. We look forward to hearing good news from ICLR 2026.

---

### Meta-Review · Area_Chair_RHqy · 2026-01-03

**Summary:**

The paper proposes FedDOR, a prototype-based heterogeneous federated learning (HFL) method that introduces two main components:
**Orthogonal Initialization (OI)** on the client side to impose a well-separated geometry over class prototypes; **Dual Orthogonal Regularization (DOR)** on the server side to maintain this geometry through inter-class separation and intra-class alignment. While the motivation (semantic degradation in prototype aggregation) is clear and the experimental results show consistent improvements across datasets, reviewers raised several important concerns that influenced the decision:

**Key Concerns**:\
***Incremental novelty over FedORGP***: Several reviewers (WHYS, FapM, SS1N, oiEP) noted that FedDOR appears to be a relatively small extension of prior work (especially FedORGP), with the main addition being orthogonal initialization. The novelty of the method—particularly the server-side mechanism—was considered limited and insufficiently distinct from existing approaches.

***Lack of theoretical analysis in the original submission***: Reviewers (FapM, SS1N, oiEP) observed that the original paper lacked formal justification for the claimed stability of the orthogonal structure and convergence guarantees.

***Insufficient clarity and presentation***: Multiple reviewers (WHYS, FapM) mentioned issues related to notation, clarity of the proposed mechanisms, and how prototype embeddings differ from classifier weights or local prototypes. This impacted the perceived soundness and readability.

***Limited discussion of costs, limitations, and hyperparameter sensitivity***: Reviewers (SS1N, oiEP) requested analysis of computational overhead, sensitivity of loss weights, and performance under extreme heterogeneity or label imbalance.

**Reviewer Concerns:**

**Addressed Concerns**:\
The authors provided a detailed and thorough rebuttal, adding valuable theoretical and empirical evidence. Notable resolutions include:

***Theoretical justification***: The authors added a rigorous Lyapunov-based analysis (Appendix A.15) establishing the local stability of the orthogonal configuration and a convergence proof (Appendix A.17) showing that FedDOR converges at a rate of O(1/T). This was a strong response to concerns about theoretical rigor (FapM, SS1N, oiEP).

***Clarified novelty and differences from FedORGP***: The rebuttal distinguishes FedDOR’s geometry-first design from FedORGP’s MLP-based prototypes with soft constraints. The authors argue convincingly that FedDOR proactively imposes and maintains orthogonal geometry rather than correcting it post hoc.

***Hyperparameter robustness***: Appendix A.10 includes heatmap-style sensitivity analyses for λ_CE and λ_align across datasets, addressing reviewer concerns (SS1N, oiEP).

***Computational overhead***: Appendix A.16 shows that both OI and server-side regularization are computationally cheap (orders of magnitude smaller than client-side training), addressing Reviewer SS1N and oiEP.

***Handling of missing classes and partial label settings***: The authors clarified that clients only send prototypes for observed classes, and additional experiments (Appendix A.11) confirm strong performance under extreme non-IID and label scarcity.

**Outstanding Concerns:**\
Despite the strong rebuttal, some substantive concerns remain:

***Limited methodological novelty***: Even with orthogonal initialization and clearly stated geometric motivations, the core architecture and training losses remain close to prior work (e.g., FedProto, FedORGP). The added theoretical depth, while valuable, does not significantly elevate the practical innovation of the method.

***Empirical gains appear modest in some settings***: In datasets like CIFAR-10, performance margins over baselines are relatively narrow, and the dependence on OI for performance calls into question whether DOR alone is impactful.

***No formal privacy guarantees***: Reviewer oiEP noted that the claim of privacy preservation is not backed by differential privacy analysis or empirical attacks, limiting the strength of this claim.

**Reviewer Scores:**

**Reviewer WHYS (Initial Score: 4)**: Raised detailed concerns about novelty, clarity, and the persistence of orthogonality. The rebuttal addressed most points, but likely not enough to raise the score significantly. Predicted Final Score: 4 (unchanged)

**Reviewer FapM (Initial Score: 2)**: Initially found the method too close to FedORGP and lacking theoretical justification. Although the rebuttal was strong, the initial low confidence in novelty likely persists. Predicted Final Score: 4

**Reviewer SS1N (Initial Score: 6)**: Raised thoughtful concerns about cost, theoretical rigor, and limitations, all of which were addressed. This reviewer may remain positive. Predicted Final Score: 6 (unchanged)

**Reviewer oiEP (Initial Score: 6)**: Requested better theoretical explanation and cost analysis, which were added. However, privacy and novelty concerns may prevent a strong endorsement. Predicted Final Score: 6

---

### Decision · Program_Chairs · 2026-01-26

Reject